# A mechanical transition from tension to buckling underlies the jigsaw puzzle shape morphogenesis of histoblasts in the *Drosophila* epidermis

**Annafrancesca Rigato[1,2], Huicheng Meng[1], Claire Chardes[2], Adam Runions[3], Faris Abouakil[1], Richard S. Smith[4], Loïc LeGoff[1] ***

**1** Aix Marseille Univ, CNRS, Centrale Marseille, Institut Fresnel UMR7249, Turing Center for Living Systems, Marseille, France, **2** Aix Marseille Univ, CNRS, IBDM UMR7288, Turing Center for Living Systems, Marseille, France, **3** Department of Computer Science, University of Calgary, Calgary, Canada, **4** John Innes Centre, Norwich Research Park, Norwich, United Kingdom

* loic.le-goff@univ-amu.fr

**Data Availability Statement:** Data are available in the file S1 Data.ods.

## Abstract

The polygonal shape of cells in proliferating epithelia is a result of the tensile forces of the cytoskeletal cortex and packing geometry set by the cell cycle. In the larval *Drosophila* epidermis, two cell populations, histoblasts and larval epithelial cells, compete for space as they grow on a limited body surface. They do so in the absence of cell divisions. We report a striking morphological transition of histoblasts during larval development, where they change from a tensed network configuration with straight cell outlines at the level of adherens junctions to a highly folded morphology. The apical surface of histoblasts shrinks while their growing adherens junctions fold, forming deep lobules. Volume increase of growing histoblasts is accommodated basally, compensating for the shrinking apical area. The folded geometry of apical junctions resembles elastic buckling, and we show that the imbalance between the shrinkage of the apical domain of histoblasts and the continuous growth of junctions triggers buckling. Our model is supported by laser dissections and optical tweezer experiments together with computer simulations. Our analysis pinpoints the ability of histoblasts to store mechanical energy to a much greater extent than most other epithelial cell types investigated so far, while retaining the ability to dissipate stress on the hours time scale. Finally, we propose a possible mechanism for size regulation of histoblast apical size through the lateral pressure of the epidermis, driven by the growth of cells on a limited surface. Buckling effectively compacts histoblasts at their apical plane and may serve to avoid physical harm to these adult epidermis precursors during larval life. Our work indicates that in growing nondividing cells, compressive forces, instead of tension, may drive cell morphology.

**Funding:** Agence Nationale de la Recherche (ANR-17-CE30-0007, ANR-18-CE13-028, ANR-22-CE42-0010, ANR-22-CE13-0039); Excellence Initiative of Aix-Marseille University - A*Midex (capostromex), a French Investissements d'Avenir programme. This project is funded by the « France 2030 » investment plan managed by the French National Research Agency (ANR-16-CONV-0001, ANR-21-ESRE-0002), and from Excellence Initiative of Aix-Marseille University - A*MIDEX. RSS was supported by a Biotechnological and Biological Sciences Research Council (BBSRC) Institute Strategic Programme Grant to the John Innes Centre (BB/X01102X/1). The funders had no role in study design, data collection and analysis, decision to publish, or preparation of the manuscript.

**Competing interests:** The authors have declared that no competing interests exist.

## Introduction

Morphogenesis proceeds through the mechanical interaction of cells in order to shape tissues. Our understanding of the cellular processes that power shape changes has considerably improved in recent year [1–4]. At the single cell scale, the role of internally generated active forces from the cytoskeleton in setting morphological changes has been well established [2,5,6]. But the shape of a cell is also determined by environmental constraints, as adhesion links cells to their neighbors and to the extracellular matrix [7]. Growth can act as a potent environmental constraint to shape cells and tissues. Spatial variations in the orientation or the rate of growth lead to mechanical prestress impinging upon cells [4,8–11]. For example, if one or a patch of cells grows more than its surroundings, the overgrowing patch will be compressed and the surroundings will be stretched [12].

Because of their biological relevance and ease of imaging, epithelia have been particularly well characterized in terms of mechanics. The combination of modeling, analysis of cell shape, and mechanical perturbations has led to the following understanding of epithelial mechanics: (i) the tissue is in a tensed state; (ii) cell growth is usually balanced over the cell cycle (cells double in volume from beginning to end of the cell cycle); and (iii) stress can be released through topological transitions such as cell neighbor exchange and oriented cell divisions [13]. With these elements, models were developed for epithelial tissues, able to capture many features of morphogenesis [14,15]. Epithelial monolayers are then described as tensed networks, formed by polygonal-shaped cells with straight borders [16,17]. Nevertheless, tensed epithelia that respect the characteristics above are only a portion of the complex scenario of morphogenesis.

Other than tensile forces, compressive forces are also important shape generators. For example, an elastic body under compressive forces can go through buckling instability [18–20], a process at play in gut vilification [21,22], in gastrulating embryos [23], in the formation of brain cortical lobules by differential growth of apposed cortical cell layers [24], or in some cell-ECM systems [25]. While these examples are taken from elasticity operating at a large scale, in this work, we find evidences that similar effects exist at the level of individual cells. In particular, we show that this happens during epidermal morphogenesis in *Drosophila*.

The epidermis of the *Drosophila* larva consists of two cell populations: the larval epithelial cells (LECs), which are large polyploid cells, and the histoblasts, which are the precursors of the adult epidermis. These two cell populations form a continuous cell monolayer [26]. Histoblasts are clustered in nests of a fixed cell number (5 to 17 cells per nest) surrounded by LECs. Growth of both histoblasts and LECs happens without cell division over a large span of larval development, from 4 h after hatching until the pupal stage [26,27]. Histoblasts do not exchange neighbors either during this period [28]. Thus, there is a complete absence of rearrangements of the cellular lattice. The growth rates of histoblasts and LECs are different, with LECs increasing in volume about 150-fold during larval life, and histoblasts 60-fold [26]. The larval body stops growing around 90 h after egg laying (h AEL) [26,29], while the epidermal cells continue to grow. Thus, this binary cellular system, where two cell populations grow and compete for space in the absence of stress-releasing topological transitions is likely to present a mechanical regime yet unexplored by other epithelial model systems.

Here, we investigate the shape of developing histoblasts. We developed a protocol for time-lapse imaging of individual cells throughout larval stages. We observed that histoblasts go through a considerable morphological transition between 90 h and 110 h AEL, changing from a tensed network configuration with straight cell outlines to a highly folded morphology of cell shapes, suggestive of compressive forces. We show that the formation of folded junctions arises from the frustrated growth of adherens junctions of histoblasts in the spatially constrained apical domain of these cells. The imbalance between growth of junctions and shrinkage of the

apical domain of histoblasts compresses the junctions and triggers a buckling instability. We also show that growth of cells on the epidermis creates a packing that affects the apical size and shape of histoblasts. We thus propose that morphogenesis of histoblasts in the *Drosophila* larva is compression-driven.

## Results

### Histoblasts undergo a dramatic morphological change during the L3 stage

To investigate the growth and morphology of histoblasts during the last larval phase, we optimized a protocol for live imaging of the larval epidermis (see Materials and methods and S1 Fig). Briefly, larvae were anesthetized with Desflurane to prevent muscle contractions and oriented to image the histoblasts. After imaging, larvae were put back in soft medium, necessary for normal growth, at 25˚C. Recovery from anesthesia takes few minutes, and larvae can then develop normally. By repeating such procedure every few hours, we could image the same cells over several hours, without affecting larval development.

In each abdominal segment (1 to 7) are 2 ventral, 2 spiracular, and 4 dorsal (2 anterior and 2 posterior) histoblast nests. We imaged the anterior dorsal nest—chosen for its larger cell number and distance from abdominal segment border where imaging is diffcult. We first imaged the adherens region, which plays an important role in epithelial morphogenesis [30,31], with an E-cadherin fluorescent protein fusion. We observed that histoblasts have straight cell borders and a polygonal shape up to about 90 h AEL. This morphology, most commonly found in epithelia, is characteristic of tensed cell networks [14]. Fig 1A–1D shows adherens junctions of the same histoblast nest at different times between 90 and 115 h AEL. After 90 h AEL, some cell junctions present local deformations (Fig 1B), which become more prominent in the following hours (Fig 1C). At the wandering stage (approximately 110 h AEL), the small wrinkles have become deep lobules (Fig 1D). These lobules persist up to pupariation, when histoblasts initiate a series of fast cell cycle under the influence of accumulated cyclin E [32]. Histoblasts from all other nests go through the same morphological transition (S2 Fig). The emergence of lobules is well encapsulated by a quantification of cell circularity, which is the normalized ratio between the cell area and its perimeter. The circularity of a perfect circle is 1, and it decreases as the overall shape is less round. In Fig 1E, we observe a systematic decrease of circularity through the transition, from 90 h to 110 h. Fluctuating wrinkles of the adherens belt, driven by stochastic bursts of acto-myosin tension, often arise in different developmental contexts [33]. The typical lifetime of these fluctuations is of the order of minutes. By contrast, the lobules we observe in the histoblasts are not dynamic, transient structures. The lobules apparent in Fig 1A–1D evolve from wrinkles to fully developed lobes over a 20-h time window. To illustrate this fundamental difference between dynamic embryonic tissues and quasi-static histoblasts, we compare fluctuations of vertices in embryonic amnioserosa cells and wandering stage larval histoblasts in S3 Fig. The nature of the trajectories are qualitatively different, as its stands that vertices in histoblasts hardly move compared to those of amnioserosa—even after removal of the global drifts present in amnioserosa due to morphogenetic flows. We measure a root mean square of the fluctuations an order of magnirue larger in amnioserosa ($0.97 +/- 0.2 \mu m$) compared to histoblasts ($0.07 +/- 0.05 \mu m$, within our measurement error). Thus, with respect to the typical time scale of cytoskeletal fluctuations (seconds to minutes), junctional lobules can be considered as quasi-static structures.

In addition to circularity, we characterized histoblast shape transition through the quantification of their perimeter (Fig 1F) and cell area (Fig 1G). Interestingly, the cell perimeter slowly increases from 60 to 70 μm between 90 and 100 h AEL, while it grows quickly up to 100 μm after 100 h AEL. On the contrary, we observe a slow decrease of cell area between 90 and 100 h

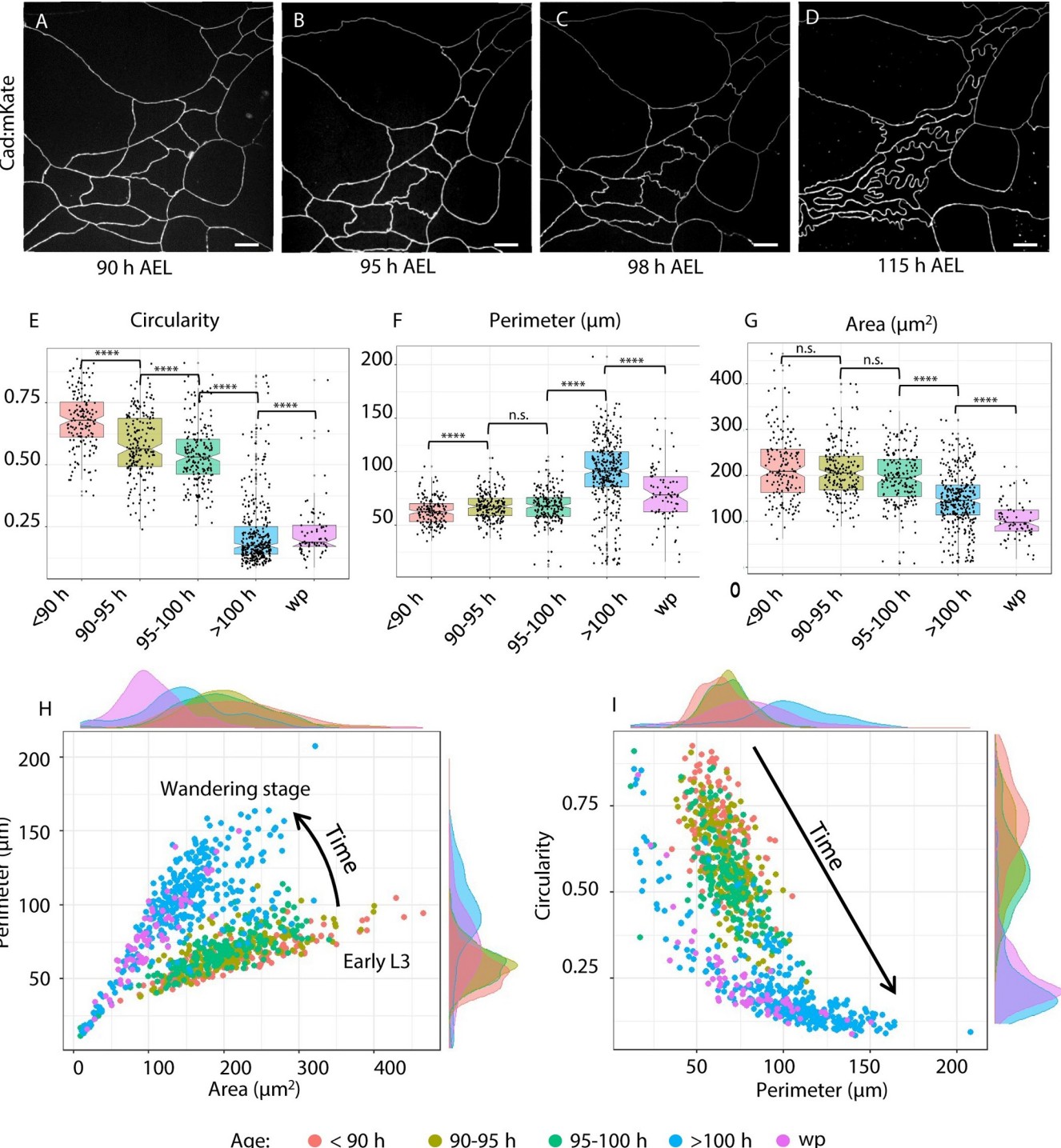

**Fig 1. Remodeling of histoblasts' junction during epidermal morphogenesis. (A-D)** Chronic imaging of cadherin junctions at different times after egg laying (AEL). At the beginning of the third instar, cell junctions are still straight. Lobules gradually develop as the larvae over the third instar from 90 h to wandering stage, reaching a maximum just before the pupariation. **(E-G)** Quantification of morphological parameters of histoblasts. Quantification of morphological parameters of histoblasts. Cell circularity. **(E)** Circularity quantifies the shape change due to the formation of lobules, with values ranging from 0 to 1. A value of one corresponds to a perfect circle. Cell perimeter is growing as the larva grows **(F)**, while cell area decreases **(G)**. Detailed $p$-values for all pairwise tests for difference can be found in Table A in S1 Text. **(H)** Scatter plot and density functions of cell perimeter vs area. The plot shows that the different ages localize in different regions of the plot, from the early L3 up to the wandering stage. **(I)** Scatter plot and density functions of cell circularity vs. area. The plot shows that the different ages localize in different regions of the plot, from the early L3 up to the wandering stage. In particular, the longer the perimeter (and the older the larva), the lower its circularity. For each time interval, the number of analyzed cells is $N = 187$, $N = 225$, $N = 186$, $N = 352$, $N = 74$. Details on

statistical analysis can be found in the Materials and methods section and Table A in S1 Text. The data underlying the graphs shown in the figure can be found in S1 Data. Scale bar = 10 μm.

AEL, and faster decrease from 200 $\mu m^2$ at 100 h AEL to 120 $\mu m^2$ after 100 h AEL until the pupal stage. As the number of histoblasts in each nest remains constant through the transition, this implies a corresponding decrease of area of the nest at the adherens plane. Statistical significance for the comparison of all distributions in Fig 1E–1G can be found in Table A in S1 Text (see section Materials and methods for more details). A decreasing area but an increasing perimeter results in a dramatic decrease of cell circularity, from 0.8 before the formation of lobules, to 0.2 when the lobules are fully formed. To fully characterize the shape of cells going through the transition, we represent them in the morphometric spaces *perimeter-area* and *circularity-perimeter* (Fig 1H and 1I). In both representations, histoblasts occupy a different region at different larval stages.

In particular, because of the opposite tendency of perimeter (increasing) and area (decreasing), cell circularity changes dramatically, and its distributions at different ages are easily distinguishable. Based on these results, we used circularity as a reference parameter for the formation of lobules in our further quantifications.

## As their apical surface shrinks, histoblasts grow basally

To our knowledge, the only available data about epidermal growth in this system was obtained by electron microscopy and estimated from 2D data by Madhavan and colleagues [26], who reported a 150-fold volume increase of LECs and 60-fold of histoblasts. While an increasing perimeter at the adherens plane seems compatible with growth of cells, the decrease in apical area is not. To further investigate this apparent inconsistency, we analysed histoblasts growth in 3 dimensions, from the beginning of lobules formation to the wandering stage. We analyzed the 3D shape of histoblasts in the course of the morphological transition by imaging their basolateral membrane with a src:GFP fusion and adherens junctions with an E-cad:mKate fusion (Fig 2). Before the lobules appear, cell borders have the same straight shape at the adherens and basal planes (Fig 2A–2C). From the orthogonal projections in Fig 2C (ZX and ZY), the distance between the basal and apical surfaces of histoblasts is about 6 μm, which is comparable to the thickness of LECs [34]. Through the transition, and while cells shrink apically, the basal side of histoblasts expands in a rounded bulb-like shape (Fig 2D–2F). As shown in Fig 2D, the nest is much larger basally than apically, and no fold is visible basally. The difference in morphology of the apical and basal sides are well apparent on the overlay of the basal and apical membranes of Fig 2G. Additionally, cell thickness is greatly increased in the course of the transition, with the apical and basal plane now being well separated, as shown from the orthogonal projections (ZX, ZY) in Fig 2F. The increase of cell thickness correlates well with the formation of lobules, as demonstrated by plotting the thickness against cell circularities (Fig 2I). The average cell thickness increases from about 6 μm to 10 μm in the course of the morphological transition (Fig 2I). As the apical area decreases while the height increases, we segmented histoblast nests after expression of a cytosolic GFP with a histoblast driver (esg-Gal4) to estimate their volume in the course of the transition. Because the cell number is constant in the histoblasts at this stage, the measurement gives us the cell-volume increase averaged over the nest. The measurements were done on individual nests, tracked over time through chronic imaging at 3 different time points. As reported in S4 Fig, cell volume increases 2-fold between 90 and 115 h AEL. Hence, histoblasts do grow during the last larval stage, despite the apical shrinkage.

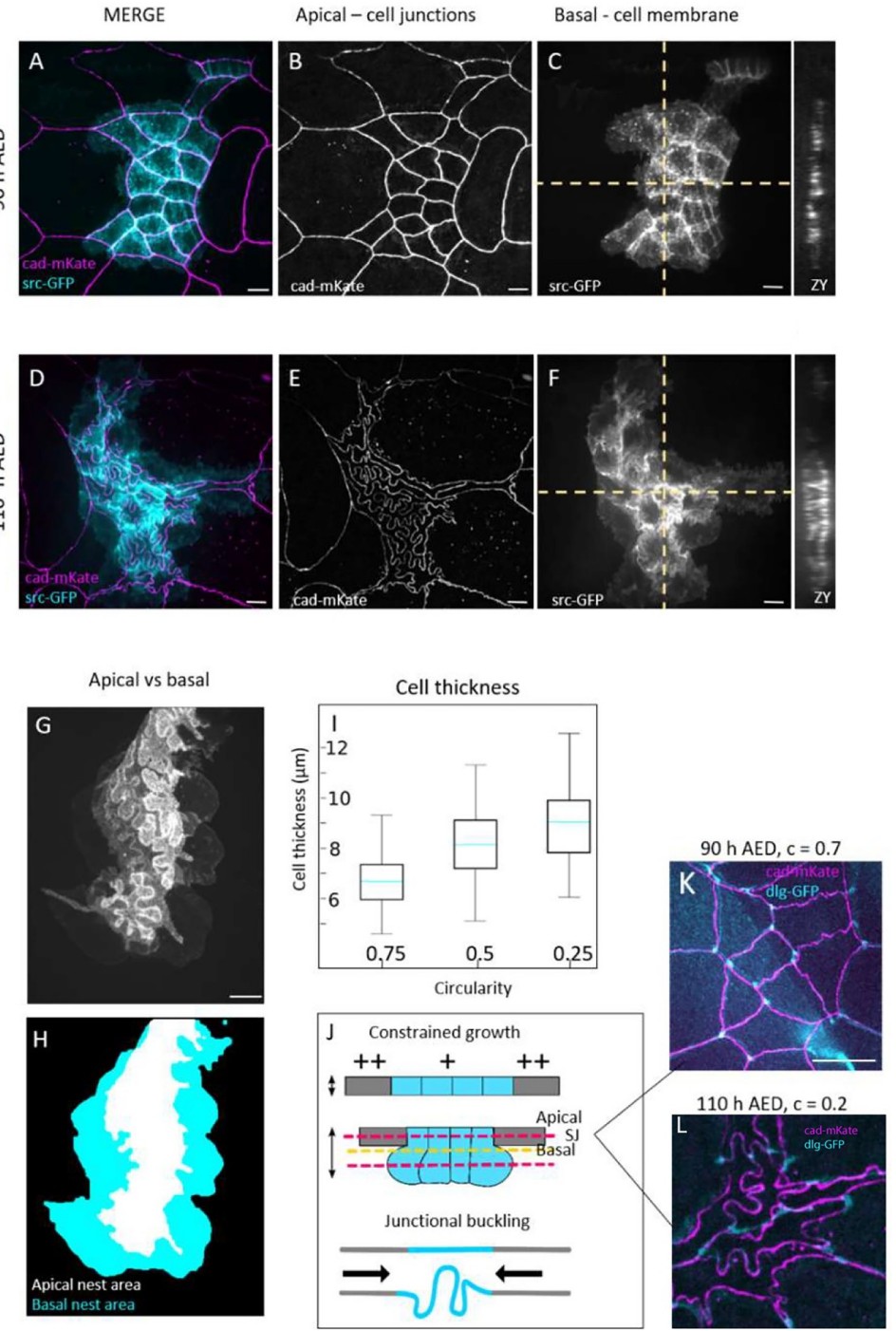

**Fig 2. 3D characterization of histoblast growth and junctional buckling.** (**A-F**) Live imaging of cadherin junctions (E-cad:mKate) and cell membrane (src:GFP) before (**A-C**) and after (**D-F**) apical junction remodeling. (**G, H**) Superposed projections of the apical and basal plane of a histoblast nest at 110 h AEL. While the apical side presents junctional lobules, the basal side is characterized by large and bulb-shaped cells. (**I**) Cell thickness plot as a function of cell circularity. Light blue line represent the means, error bars the standard deviation. Thirty different nests were analyzed, from 10 larvae. (**J**) Schematics of cell growth below the apical surface as junction remodeling occurs. At the top, section of histoblast and LECs showing how histoblast develop below the epithelial surface. Red dashed lines represent the apical and largest basal plane corresponding to images A-F. The yellow dashed line represents the plane of septate junctions, corresponding to images K and L. (**K, L**) Merged projection of apical junctions and septate junctions at 90 h AEL (**K**) and 110 AEL (**L**). Before junctional buckling, apical and septate junctions have similar, superimposing structures. When apical junctions fold, the deformation is partially lost at the plane of septate junctions.

White dashed circle: example of straight septate junction. Yellow dashed circle: example of septate junction not following the corresponding apical junction. The data underlying the graphs shown in the figure can be found in S1 Data. Scale bar = 10 μm.

As shown in Fig 2G, histoblasts are round-shaped at the basal side. Hence, the lobules observed at the level of adherens junctions are not a characteristic of the entire lateral cell membrane. To investigate whether the lobules are only localized at the apical plane and lost immediately below, or whether they are gradually lost, we imaged septate junctions with a disc large protein fusion (dlg:GFP) together with E-cadherins (cad:mkate). Septate junctions are localized just below adherens junctions. Before the morphological transition, septate junctions have the same shape as adherens junctions—their projections superpose (Fig 2K). When lobules appear, septate junctions follow only partially the shape of adherens junctions despite being localised very close to them (Fig 2L). Thus cell border at the level of septate junctions are less folded than at the level of the adherens junctions but more than at the basal level.

In summary, over the time window from 90 h to 110 h AEL, histoblasts increase their volume 2-fold while their adherens junctional material also increase by a factor of 1.7. However, in that same time window, their apical area decreases. The volume increases at the basal side of cells. We formulated a working hypothesis summarized in Fig 2H, whereby junctional folding in histoblasts is generated by the continuous growth of junctions, which is not balanced by a concomitant expansion of the cell population at their adherens plane, like in most epithelia studied currently. Growth of a slender elastic structure in a constrained environment leads to its compression and eventual buckling [35]. We thus propose that the lateral constraints that lead to shrinking of the apical surface of histoblasts could also influence the mechanical buckling of the growing junctions.

## Manipulation of junctional growth affects junctional folding

In our hypothesis, histoblasts' buckling is the result of the concomitant growth of their junctions and the shrinkage of the apical domain of the histoblast nest. To test the contribution of junctional growth, we specifically altered the growth of adherens junctions in histoblasts by genetic means. By reducing the available junctional material, we expect a reduction of folding. First, we impaired the activity of Rab11, known for its role in cadherin recycling.

Rab11 is responsible for the transport of newly synthesized cadherin as well as recycled cadherin and other proteins to the cell junctions [36]. We induced the overexpression of a Rab11-dominant negative (Rab11-DN) specifically in the histoblasts. In Fig 3A and 3B, wild-type (WT) histoblasts have long and folded junctions, while Rab11-DN have shorter and straighter junctions. We measured a circularity of 0.74 for Rab11-DN against 0.17 in WT ($p$-value = $4.10^{-38}$), an average junction length of 30 $\mu m$ in Rab11-DN against 100 $\mu m$ in WT, ($p$-value = $3.10^{-32}$).

Next, we directly impaired Cadherin levels by silencing its gene (*shotgun*) with RNA interference. We expressed Cad-RNAi in the histoblasts (esg > Cad-RNAi) of larvae carrying src: GFP as a membrane marker to compensate for the absence of an adherens marker. While junctional folds are visible in WT membrane-labeled histoblasts (Fig 3D), cadherin-depleted junctions do not develop folds at the wandering stage and have polygonal-shaped cells (Fig 3E). The measurements of cell morphological parameters (Fig 3G–3I) reveal no significant change in cell area, but a 2-fold decrease of cell perimeter (median value 52$\mu m$ versus 100 μm, $p$-value $10^{-19}$ according to Mann–Whitney test), with a consequent increase of circularity from 0.2 to 0.7 ($p$-value = $5.10^{-25}$, Mann–Whitney test). The scatter and density plots in Fig 3F confirm that Cad-RNAi and Rab11-DN (resp. green and cyan) cells are found in a different

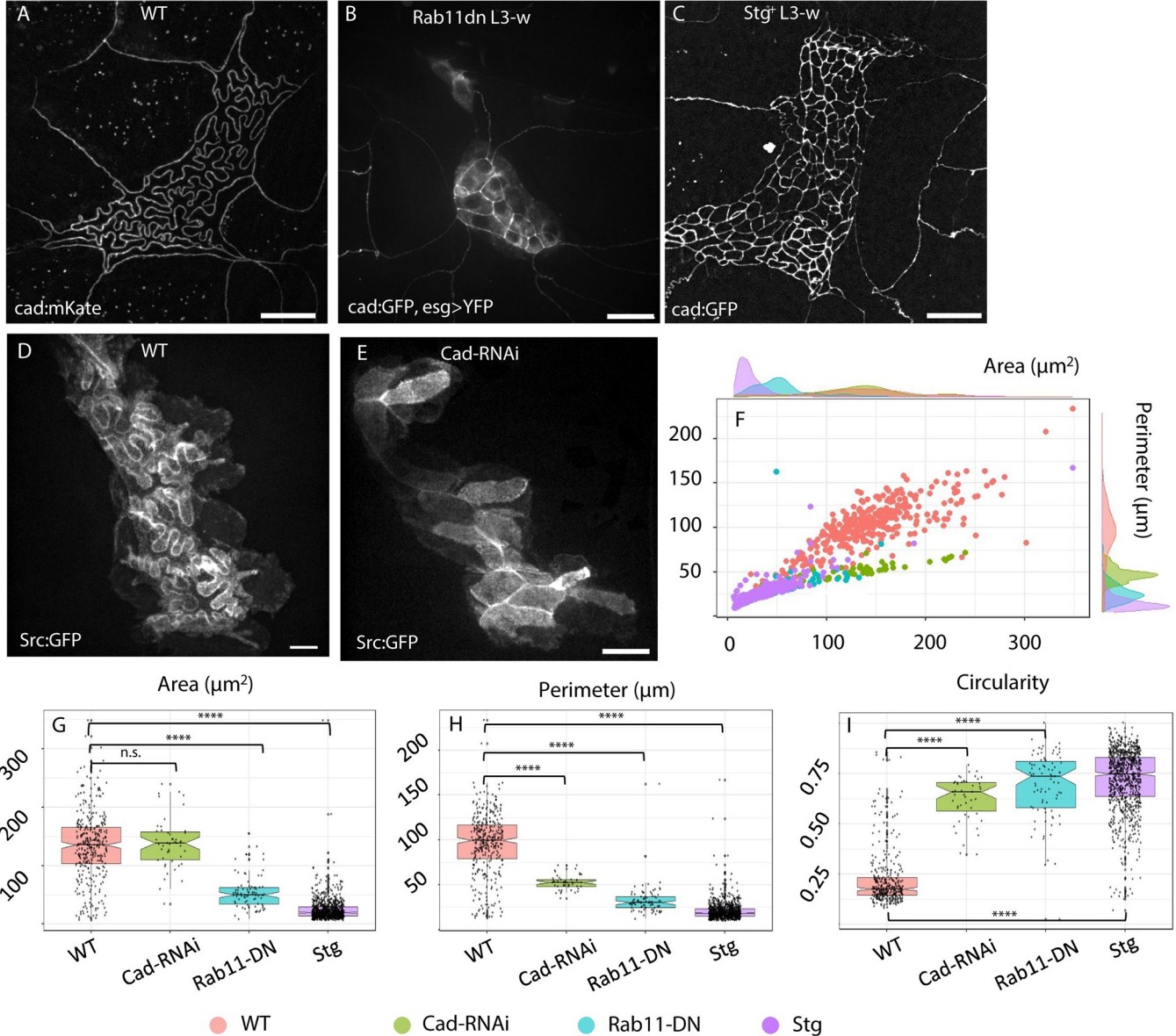

**Fig 3. Genetic perturbation of histoblasts junctions and cell cycle. (A)** Live image of junctions (E-cad:mKate) of WT *Drosophila* larva, at wandering stage (110 h AEL). (**B**) Live image of junctions (E-cad:GFP) of *Drosophila* larva, at wandering stage (110 h AEL), upon E-cadherin recycling impairment in histoblasts (esg>Rab11-DN). Histoblasts show shorter, straight junctions and reduced apical area. The diffused cytosolic signal is due to a YFP linked to Rab11-DN, partially emitting in the GFP spectrum. (**C**) Same, upon expression in histoblast of Cdc25/Stg (esg>Stg). Histoblasts are more numerous, smaller, buckling is abolished, and the nest is larger. (**D**) Live imaging of histoblast membranes (esg>src:GFP) in a WT larva, at the wandering stage. The z-projection shows both the apical buckled junctions, and the basal round shape of histoblasts. (**E**) Same as D, upon down-regulation of E-cadherin in histoblasts (esg>Cad-RNAi). Histoblasts' junctions appear straight, and the basal side as round and wide as in the WT larva. (**F-I**) Morphometric quantifications. (**E**) Scatter plot and density plot of perimeter vs. area of WT (pink), esg>Rab11-DN (green) and esg>Stg (blue) histoblasts. (**F-H**) Box plots of essential morphological descriptors: perimeter (**F**), cell area (**G**), and circularity (**H**). Details on statistical analysis can be found in the Materials and methods section and Table C in S1 Text. Scale bar = 20 μm. WT = wild type, N cells = 352 (WT), 49 (Cad-RNAi), 82 (Rab11-DN), 880 (Stg+). Average cell number per nest = 15 ± 2 (WT, ±STD), 12±2 (Rab11-DN), 108±22 (Stg+), 14±2 (Cad-RNAi). The data underlying the graphs shown in the figure can be found in S1 Data.

region than WT cells (pink) in the *perimeter-area* morphometric space. Combined, these experiments provide evidence that the morphological transition that we observe depends on the addition of junctional material to the adherens belt of histoblasts.

## Reintroduction of cell division abolishes junctional folding

In the context of a buckling instability, the length of junctions is an important intrinsic factor that controls the critical compressive load at which buckling proceeds. Indeed, for an elastic beam, this critical load scales as $\sim L^{-2}$, where $L$ is the characteristic length of the beam [18]. By depleting junctions from their E-cadherin pool, the previous cad-RNAi and Rab11-DN experiments may have changed the material properties of junctions. One means to change the length of junctions, without altering the rate of junctional material addition, is to reintroduce cell divisions in histoblasts. The net result is a denser network of shorter junctions, while maintaining the external mechanical constraints on the nest unchanged, as well as the material properties of junctions such as the density in E-cadherins or their elasticity.

We forced histoblasts to divide by overexpressing the mitotic controller cdc25 –*string* (*stg*) in *Drosophila* [37]. This perturbation only impacts the cell cycle and not growth [38]. As a consequence, histoblasts were more numerous in the histoblasts nests (around 100). As shown in Fig 3D, individual histoblasts are consequently smaller and have straight junctions. This experiment further supports the model whereby buckling of the junctions proceeds because of their excessive lengthening while the apical area is constrained. It also highlights the essential role played by stalling of the cell cycle. Would the histoblasts divide like, for example, cells of the imaginal discs, the deep junctional lobules would not occur.

Besides the direct effect on cell junctions, the whole histoblast nest is also larger than in the control (Fig 3C). A possible interpretation for this experiment is that the shortened length of junctions prevents the onset of buckling and thus improves their ability to withstand mechanical stress. As the junctions are less prone to buckling, the overall nest becomes stiffer and gets squeezed to a lesser degree. Thus, forcing cell divisions in histoblasts abolishes buckling and reduces compaction of the histoblast nest.

## Histoblasts in the larval epidermis are not under tension and have elastic material properties

According to our buckling working hypothesis, the growing junctions are compressed by the limited apical area of histoblasts. This implies that folding is not a tension-driven morphology. To test this, we ablated both LECs and histoblast junctions during the morphological transition (around 95 h AEL). We performed laser dissection on cad:GFP with a custom-built setup (see Materials and methods). First, we tested our system on a control, tensed epithelium—the histoblasts after complete replacement of LECs by histoblasts in the pupal stage. When the adult epidermis was ablated, relaxation of the cut junctions was observed, as well as a shape change in the neighboring cells (Fig 4A, quantification in Fig 4D). Similar relaxations were observed in the wing imaginal discs, another tissue known to be under tension [12,14,39,40]. Instead, when histoblasts were ablated at the larval stage, no relaxation was observed (Fig 4B and 4D). The small relaxation observed at the onset of ablation (0 to 2 s, green curve, Fig 4D) corresponds to the disappearance of the cad:GFP signal when the hole is generated. Severed junctions in LECs, close to histoblasts and at the same larval stage, relaxed with an intermediate behavior. These cells are more tensed than histoblasts, but not as tensed as in the pupal epidermis. These results confirm that histoblast morphology at the end of the larval stage is not driven by a tension-based mechanism. Importantly, these observations also demand a refinement of the buckling hypothesis. If the morphology of the junction were solely driven by buckling of an elastic body (the junction) under compressive forces, we should observe a straightening of the junctions upon dissection. Since this did not occur, our data suggest that some remodeling must occur to stabilized buckled junctions and dissipate the mechanical

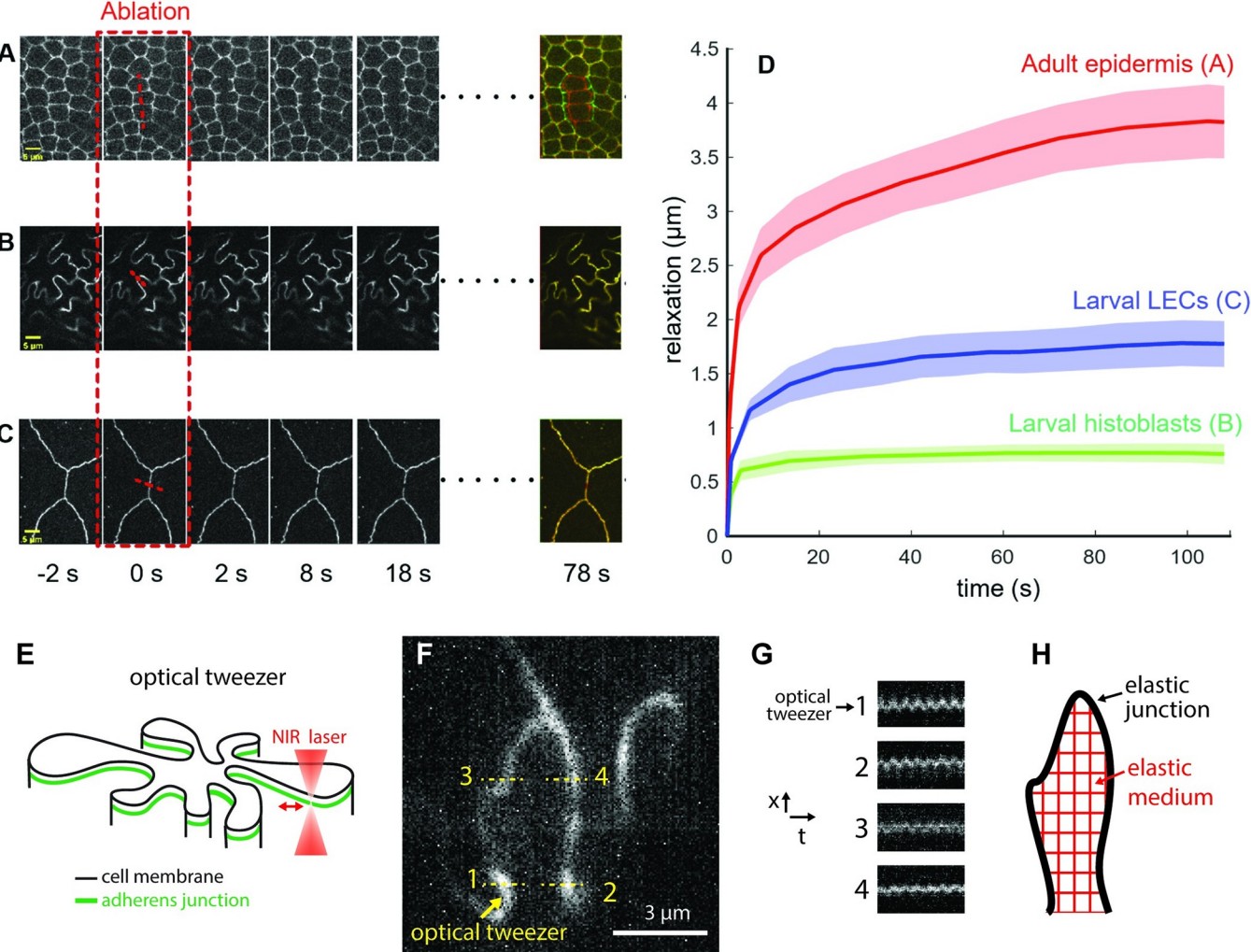

**Fig 4. Probing mechanics of histoblasts.** (**A**) Laser dissection of pupal epidermis. (**B**) Laser dissection of histoblasts in a 95-h AEL larva. (**C**) Laser dissection of LECs in a 95-h AEL larva, close to histoblasts. In A, B, and C, the last panel on the right represent the merge of the tissues before and 78 s after ablation. (**D**) Quantification of relaxation curve for the 3 conditions in A (red), B (green), and C (blue). Each curve is the average of 5 replicates. Error bars represent standard error. (**E**) Schematic of an optical tweezer experiments. A near-infrared laser is used to mechanically stimulate a junction in a point-wise fashion. (**F**) Image of a junction where stimulation proceeds at point 1 and displacements are probed at points 1, 2, 3, and 4. (**G**) Corresponding kymographs. Point 2 displays the strongest oscillation in phase with the stimulus. (**H**) Observation align with a schematic where different points of the elastic junction are interconnected by an elastic cell medium. The data underlying the graphs shown in the figure can be found in S1 Data.

stress. This could proceed through a change in the structure or composition of actomyosin in the junctions.

If histoblast junctions are not under tension, does this mean that they are soft structures, analogous to a slack and flexible rope? The absence of fluctuations in the nest during 30 min of live imaging (S3 Fig) argues against this. Had junctions been extremely flexible, one would have expected some degree of junctional movement associated with the various active processes at play in the cell. It is not clear, however, what the amplitude of these active forces are in histoblasts and thus how stiff the junctions are owing to the observed movement. We thus further investigated this by imposing an oscillatory mechanical point stimulus on junctions using optical tweezers (schematic in Fig 4E)—taking advantage of the possibility to tweeze junctions with no "handle" microbead [41]. Strikingly, the mechanical perturbation can

propagate over quite long distances. As visible in S1 Movie, a region as large as $6\mu m$ around the point of stimulation appears to move coherently, following the sinusoidal stimulus. Kymographs on selected points confirm this qualitative assessment (Fig 4F and 4G). Points 1, 2, 3, and 4 in Fig 4F and 4G display different degrees of oscillations in phase with the stimulus. Some of these points are located on the opposite side of the cell. By comparison, when we stimulated individual junctions in the germ band of early *Drosophila* embryos, the oscillation did not propagate, even to nearby junctions (S2 Movie). In S5 Fig, kymographs displacements on an opposite junction of the same cell and at abutting vertices did not display any oscillation. Thus, while embryonic tissues display a significant amount of dissipation in their mechanical behavior, as reported in Clément and colleagues' study [42], histoblasts seem to be driven by an elastic behavior characterized by storage of mechanical energy.

Across the different locations analyzed in Fig 4E and 4F, point 2 is the one that displays the strongest oscillation, reflecting the strongest coupling to the point of stimulation. This point is the closest to the stimulation point in Euclidean distance but the furthest away in terms of curvilinear distance along junctions. In fact, point 2 is not part of the same cell–cell interface. This suggests that different parts of cell junctions are elastically connected, at least partially, through the cell medium, most likely the apical cortex (schematic in Fig 4H).

## Computer simulations of buckling junctions

It stems from our analysis that junctions are elastic structures, which evolve quasi-statically as they grow. A first qualitative understanding for junctional buckling is rooted in the mechanics of elastic beams under compression [18]. As junctions grow, they are compressed because the apical domain of histoblast remains narrow—and even shrinks. Note that shrinkage of the apical surface is not a strict requirement. As long as growth of junctions is not properly balanced by a corresponding increase in apical surface, they will experience compression and will be prone to buckling.

Two observations, however, discriminate lobules formation in histoblasts from the simple buckling of a beam (Fig 5A and 5B). First, an elastic beam under compression buckles through its lowest mode of deformation (Fig 5A, top), while histoblast junctions deform through high modes—several undulations are often observed for a single junction (Fig 5B, top). Second, if a buckling beam is suddenly cut at one point, it should relax to its resting shape (Fig 5A, bottom). Instead, histoblast junctions hardly relax after laser ablation (Fig 5B, bottom), as shown in Fig 4.

High mode buckling can easily be rationalized and should, in fact, be the expected outcome based on simple elastic considerations. Indeed, lateral junctions are not mechanically isolated structures. They are embedded in their environment through connections to the cytoskeleton and the apical cell domain. While a molecular description of such a connection is beyond the scope of this study, our optical tweezer experiments demonstrated the interconnection of large domains in histoblast, most likely transmitted through the cell cortex. On simple elastic terms, one can view the junction as a slender beam connected to an elastic substrate or "foundation." Such an elastic foundation precludes low mode buckling because their large amplitude are too costly in elastic energy. High-mode buckling has, for example, been observed in compressed microtubules, when coupled to the surrounding cytoskeleton or to immobilized molecular motors in a motility assay [43,44]. At a larger scale, high mode buckling is also observed in morphological instabilities of soft materials [45].

The absence of relaxation upon ablation suggests that a process is at play in the course of the morphological transition to dissipates the compressive stress born by junctions. The most likely underlying mechanism would be a reorganization of the cytoskeletal cortex. It is not

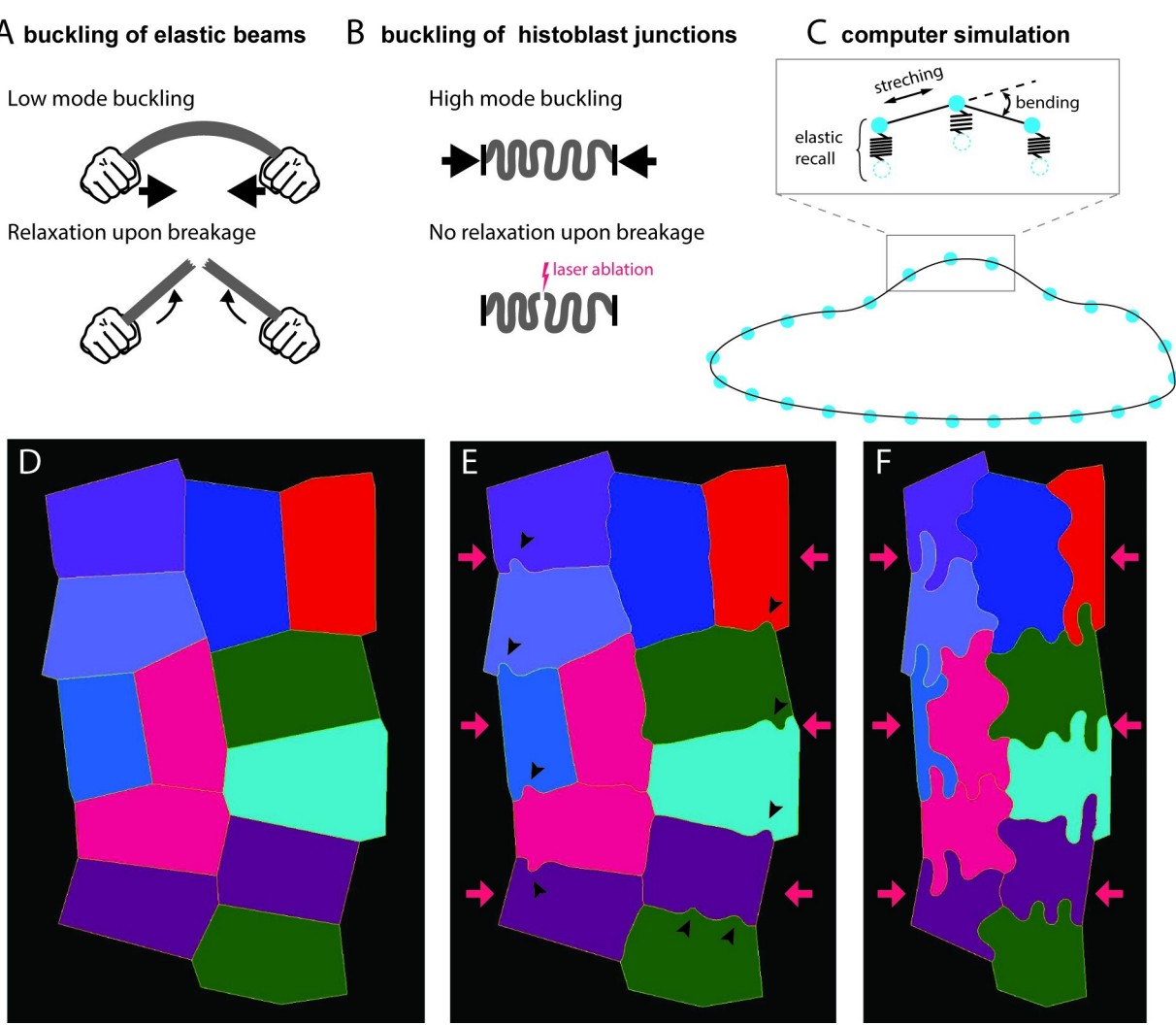

**Fig 5. Computer simulation of buckling. (A, B)** Some features of histoblast buckling are incompatible with simple elastic beam buckling. (**A, top**) An elastic beam under compressive stress at the extremities buckles through low mode deformation. (**A, bottom**) The stored bending energy is released upon breaking the beam, which results in a recoil of the beam. (**B, top**) Histoblasts buckle at small wavelengths (high mode). (**B, bottom**) Upon laser dissection, junctions hardly relax. (**C**) Computer simulations of histoblast mechanics. Cell boundaries are represented as point masses (vertices) connected by springs (edges). The elastic energy of the boundary includes stretching and bending terms. Connection to the elastic environment is simulated with springs that pull vertices back to their position. (**D-F**) Snapshots of the computer simulation of an histoblast nest going through the morphological transition. In the initial configuration, cells are polygonal (**D**). Early on in the transition, isolated lobules appear along the cell–cell interfaces (arrowheads in E). Lobules have fully developed in the final state.

clear at this point whether this dissipation is triggered by the high stress experienced by buckled junctions—a characteristic of plastic processes—or whether time plays a critical role. In any case, to the best of our knowledge, no work has addressed the possibility of a mechanical buckling process in the presence of a stress-dissipating adaptation to stabilize previous deformations.

We designed a computer simulation of tissue mechanics that encapsulates the essential elements at play in the course of the morphological transition. This simulation was implemented in MorphoDynamX using a similar approach to that published in Sapala and colleagues' study [46], which is well adapted to describe cell assemblies under either a tensile or compressive stress. Cell boundaries are represented as point masses (vertices) connected by springs (edges,

Fig 5C). One cell boundary is constituted of many such edges (approximately 50 to 100) to provide a continuum description. The elastic energy of the boundaries includes stretching and bending terms (inset in Fig 5C). At each iteration loop, the length of edges and the outer boundaries of the entire nest are assigned new values that reflect the growth of cell–cell junctions and shrinkage of the nest. Mechanical equilibrium is then found using the backward Euler implicit solver. The simulation includes a plastic deformation leading to stress dissipation at each iteration loop and an elastic recall that pulls vertices back to their position, to account for the elastic embedding of the cell–cell junction in the surrounding cell medium—thus introducing the elastic foundation. We start the simulation with a polygonal shape (Fig 5D). The shrinkage of the nest and lengthening of the cell–cell junctions is then induced. Lobules, associated with buckling, first appear in an isolated fashion along cell junctions (black arrowheads in Fig 5E). Note that similar isolated irregularities are observed at the onset of the transition in live samples (Fig 1B). As the simulation proceeds, the lobules become more pronounced, eventually giving rise to a final state highly reminiscent of the experimental observations (8F).

These simulations show that a simple mechanical model of junction growth in a constrained shrinking space leads to buckled junctions very reminiscent of the ones we observe experimentally. Notably, a nontrivial outcomes is that elastic buckling can proceed even when a slow process allows to dissipates mechanical stress through a plastic process.

## Junctional buckling is accompanied by a partial remodeling of the cell cytoskeleton from apical junctions

The acto-myosin network plays a prominent role in setting the mechanical state of apical junctions [30,31,47]. We assessed its rearrangement in the course of the folding transition.

We first imaged Myosin II, through a GFP fusion of its regulatory light chain (*Drosophila* spaghetti squash, sqh:GFP) together with E-cadherin (cad:mkte). As can be seen in Fig 6, MyoII localizes at histoblasts junctions before the transition (Fig 6A and 6B), but not as much after (Fig 6C and 6D). While E-cadherin remains junctional once lobules have emerged (Fig 6C), the Myosin II junctional enrichment is partially lost (Fig 6D). Quantification of Myosin II levels shows a gradual decreases of junctional enrichment as junctions fold (i.e., as circularity decreases) (Fig 6E). Thus, junctional folding is accompanied by a gradual reduction of Myosin II enrichment at cell junctions.

We also observed actin through the morphological transition using an affimer-GFP fusion (af:GFP [48]; see Materials and methods) before and after the transition (S6 Fig). We observed a reduction of the junctional pool of actin, in a similar way to Myosin II, albeit to a lesser degree (S6 Fig). Interestingly, from the orthogonal projections of S6B and S6D Fig actin becomes visible along tricellular interfaces in the course of the transition, connecting the apical and basal plane.

The measurements on Myosin II and Actin thus suggest a moderate loss of cytoskeletal components from junctions. This goes in the direction of a reduction in junctional tension and could add to the mechanism aforementioned of compression of growing junctions in a constrained environment. A reduction in Myosin levels is also usually associated with a reduction in stiffness [49], which could potentially contribute to the buckling process by lowering the critical force needed to buckle junctions. This critical force is proportional to bending stiffness in the case of an elastic beam [18]. A movement of cytoskeletal elements to the medio-apical cortex could also contribute to its stiffening in support of of the elastic foundation hypothesis and our observation that forces imposed with an optical tweezers on a junction can be transmitted through the cell medium. Lastly, the enrichment in actin along the the

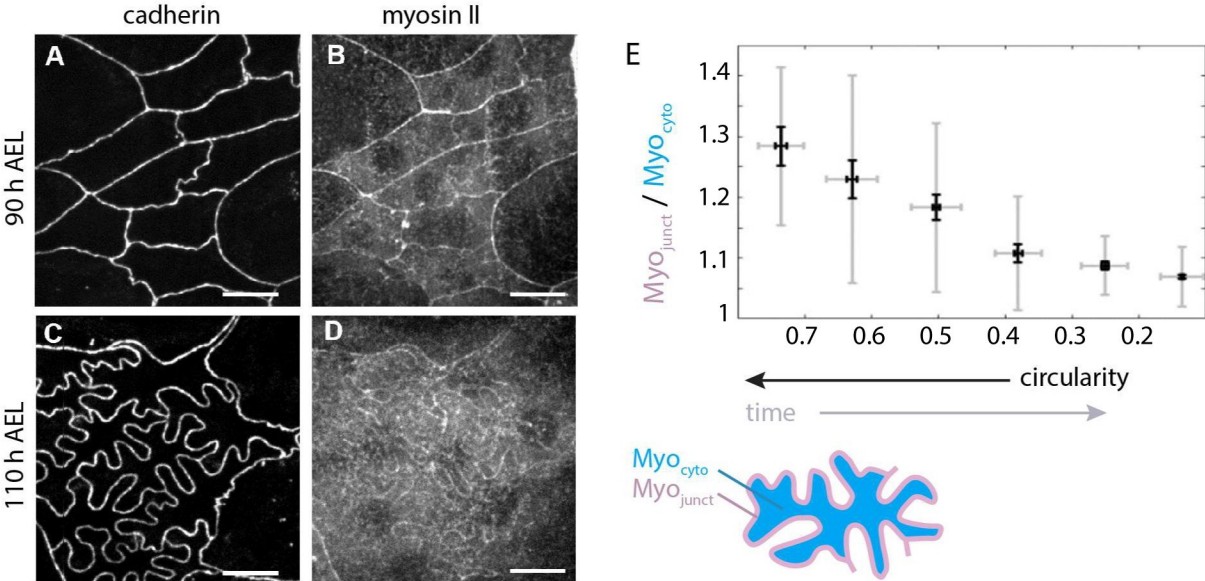

**Fig 6. Localization of Myosin II in histoblasts before and after apical junction remodeling.** Live imaging of E-cadherin (E-cad:mKate) and myosin II (MyoII:GFP) in the junctional plane of histoblasts at 90 h AEL (**A, B**) and 110 h AEL (**C, D**). Scale bar = 10 μm. (**E**) Plot of relative amount of junctional myosin as a function of circularity, calculated as the ratio of junctional ($Myo_{junct}$) over cytosolic ($Myo_{cyto}$) pools, as represeted in the schematic. Note that the axis of circularity has been inversed to reflect temporality. In the box plot, the horizontal bar represents the median for each bean, the shadowed areas the confidence interval of 0.05, the diamonds correspond to the mean value for each bin and the yellow circles are single data points. Pearson's correlation coefficient calculated on all points is 0.59 with $p$-value $p\sim10^{-32}$. $t$ Test comparisons for the junctional enrichment of the first and last point gives $p$-values of $p\sim10^{-33}$. For each bin, $N$ = 17, 29, 45, 39, 50, 199. A total of 29 histoblast nests with about 15 cells/nest were analyzed. The data underlying the graphs shown in the figure can be found in S1 Data.

tricellular interfaces could support the redistribution of the growing histoblasts content towards the basal side of the epithelium (see Fig 2). Overall, these observations show that there is a slow dynamics of the cytoskeleton at junctions of histoblasts. Although we do not observe fast dynamics on minute time scales like in the amnioserosa, for example [33], slow rearrangements on the hour times scale are present and could play a role in the plastic process leading to stress relaxation.

## A mechanical tug of war with LECs sets the apical size of histoblasts

If buckling arises from the growth of adherens junction in a spatially constrained environment, what mechanism underlies this spatial restriction? In a simple mechanical representation, a lateral pressure builds up as the 2 cell populations, LECs and histoblasts, grow on the limited surface of the epidermis. While both cell types are subjected to this 2D pressure, the way they are affected by it is different and depends on their physical properties. The size of the histoblast nest is dependent on how histoblasts withstand this pressure. Importantly, LECs remain squamous throughout the transition; thus, only a thin apical domain of histoblast in contact with LECs can experience this lateral pressure. To test the hypothesis of a 2D pressure induced by growth, we perturbed the metabolism of LECs using the driver e22c-Gal4, which is specific to LECs starting from L3 stage. While histoblasts are not targeted by the perturbation, they should experience it through a change in the lateral pressure, if our working hypothesis is correct. First, we impaired the insulin pathway specifically in LECs through the overexpression of a dominant-negative form of the *Drosophila* insulin receptor (InR-DN) [34,50]. The expression of the impaired insulin receptor was limited to the L3 phase by using a temperature-controlled expression (Gal80-ts). This resulted in a selective growth reduction of the LECs,

starting from around 90 h AEL. As a result of the perturbation, the apical area of LECs was significantly reduced at the wandering stage with respect to the WT, going from approximately 1,500 $\mu m^2$ to approximately 1,000 $\mu m^2$ (Fig 7D, *p*-value ~0.01, representative image in S8 Fig). As we hypothesized, this perturbation also impacted histoblasts, the apical area of which (at the adherens plane) changed significantly from 135 $\mu m^2$ to 189 $\mu m^2$ (representative image in in Fig 7B, quantifications in Fig 7E and 7F, *p*-value ~0.02).

Inversely, we increased LECs growth by expressing TSC1-RNAi in LEcs. TSC1 is a tumor suppressor that controls growth by antagonizing the insulin and TOR pathways. Silencing of TSC1 results in cell overgrowth [51,52]. This time, the apical area of LECs at the wandering stage was significantly increased by the perturbation with respect to the WT, going from 1,500 $\mu m^2$ to 1,860 $\mu m^2$ (Fig 7D, *p*-value ~0.01, representative image in S8 Fig). Again, the perturbation also impacted histoblasts nonautonomously, but this time decreasing their apical area (at the adherens plane) from 135 $\mu m^2$ to 87 $\mu m^2$ median values (representative image in Fig 7C, quantifications in Fig 7E and 7F, *p*-value ~$4 \cdot 10^{-6}$).

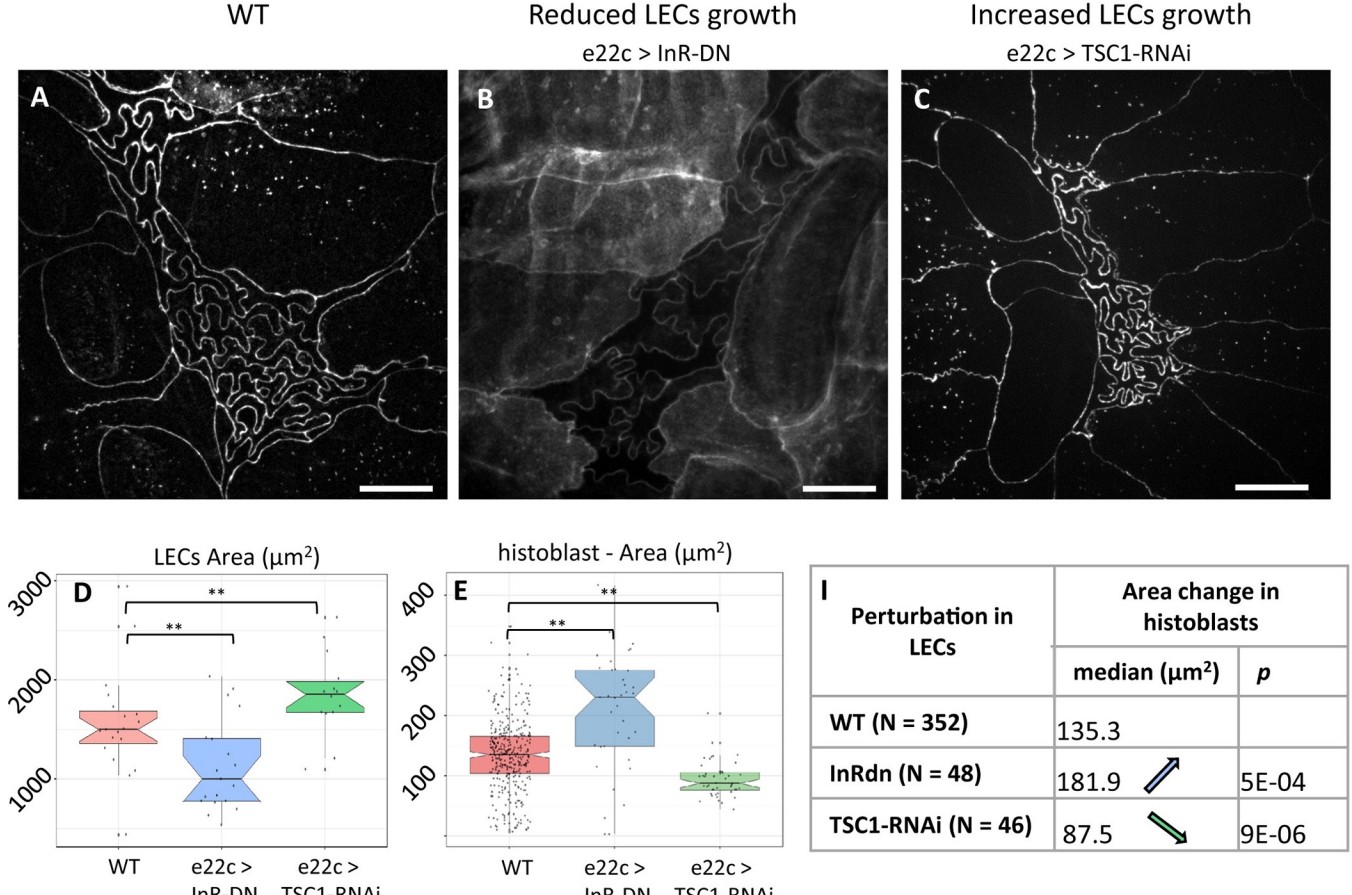

**Fig 7. Growth modifications in LECs affects spatial restrictions on histoblast nests. (A)** Live imaging (cad:mKate) of WT *Drosophila* epidermis (110 h AEL). (**B**) Live imaging (cad:GFP) of the epidermis (110 h AEL) when LECs growth was reduced through insulin pathway alteration (e22c>UAS-InR-DN, note that the membrane reporter src:GFP was coexpressed with InR-DN). (**C**) Live imaging (cad:mKate) of larval epidermis after increase of LECs growth through expression of TSC1-RNAi (e22c>TSC1-RNAi). Scale bar = 10 µm. (**D, E**) Qualification of the apical area of LECs (**D**) and histoblasts (**E**) upon the aforementioned alterations of LECs growth. (**F**) Table of histoblast area changes upon the aforementioned alteration of LECs growth. Further details on statistical analysis can be found in Tables D and E in S1 Text. Average cell number per nest = 16 1 (WT, STD), 16 1 (e22c>InR-DN), 14 22 (e22c>InR-DN). The data underlying the graphs shown in the figure can be found in S1 Data.

When altering insulin signaling in LECs (e22c>InR-DN), we observed that histoblasts' junctions are less folded than in the WT. We measure a reduction in histoblast circularity, with a maximized effect at the very beginning of the pupal stage, from 0.17 to 0.20 (S7 Fig). In the case of e22c>TSC1-RNAi, the circularity measurement was hampered by segmentation errors as opposite cell junctions often touched each other. This precludes the identification of individual cells. However, we were able to measure the junction straightness, which is the ratio of the end-to-end distance of a junction over its curvilinear distance. This measurements show a significant reduction in straightness from 0.67 (median) in the WT to 0.57 in e22c>TSC1-RNAi ($p = 0.004$). It is interesting to notice, that the total size occupied by the nest is about $2,400 \pm \mathbf{400} \mu m^2$ for the WT, i.e., bigger than a single LEC, while in TSC1-RNAi larvae, a full histoblast nest occupies about $1,600 \pm 300 \mu m^2$, which is about the size of one LEC. Hence, the available space for histoblasts is reduced.

In these experiments, we did not target histoblasts with the genetic perturbations. Only LECs were subjected to altered growth. The increased or decreased apical area experienced by histoblasts then does not result from a cell-intrinsic mechanism, but rather from a change in the growth-related lateral pressure. Lastly, we propose that experiments in which we forced cell divisions in histoblasts (Fig 3C, 3H, and 3I) support as well the mechanical hypothesis. Shorter and numerous cell junctions do not buckle in esg>stg nests. This geometry then grants the nest a great stiffness, which could explain the larger apical domain in the nest observed in Fig 3D.

## Discussion

We have presented a novel case of epithelial morphogenesis in which some of the canonical rules for epithelial development are no longer valid. In recent years, an emphasis has been put on the importance of junctional tension in shaping epithelial cells [14,53,54], and the release of mechanical stress through topological transitions [4,5,55]. Instead, the larval epidermis of *Drosophila* is formed by 2 cell populations that are growing but not dividing and not exchanging neighbors. LECs and histoblasts form a continuous cell layer that is initially homogeneously thin (a few μm), with LECs and histoblasts sharing similar polygonal morphologies but different sizes. We observe that histoblasts go through a morphological transition in which their junctions in the adherens plane evolve into a deeply folded shape (Fig 1), very different from the tensed, straight cell interfaces usually found in epithelia. Such an unusual shape, which we named junctional buckling, is stable over several hours. We formulated a descriptive model for junctional buckling (Fig 8). In our model, the phenomenological observation that the apical surface of the histoblast nest shrinks, while cell junctions grow, implies the compression-induced buckling of cell junctions and, thus, the formation of folds (Fig 8A). Our mechanical characterization of histoblasts using ablations and optical tweezers support the hypothesis of junctions, which are elastic and interconnected through the cell medium. However, buckling of junctions in histoblasts differs from pure elastic buckling in that some internal rearrangements must proceed quasi-statically to dissipate stress such that, when severed with a laser, the junction hardly relaxes. Computer simulations support the physical plausibility of this mechanism (Fig 5). Further investigations on the subtle balance between stress-releasing plasticity and stress-generating compression is an interesting roadmap for future studies—trying to address how the residual compression can still drive buckling. This should stimulate the development of modeling strategies that can describe cells in nontensed configurations [46,56].

We propose that the apical size of histoblasts, which plays an important role in the morphological transition, may stem at least partially from a mechanical tug of war between the 2 cell populations in the epidermis (Fig 8B). As cells grow, compressive stress builds up in the

### A   cell shape in the histoblast nest

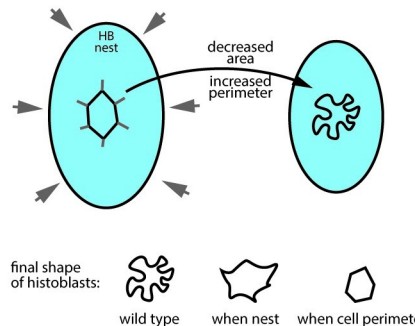

### B   mechanical tug of war in the larval epidermis

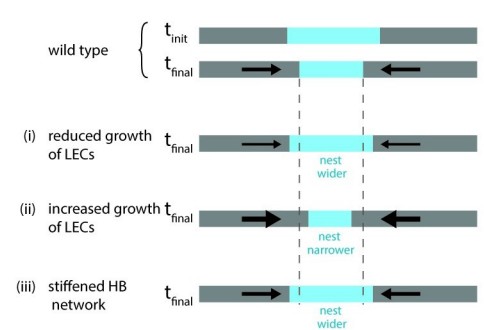

**Fig 8. Junctional buckling model. (A)** The shape of individual histoblast results from the balance between the reduction of the available apical surface and the increase of junctional material as the cell grow. The constrained growth of junctions makes them buckle. **(B, top)** A mechanical tug of war proceeds on the surface of the *Drosophila* larva between histoblasts and LECs. As pressure increases, due to cell growth on a limited surface, the histoblast nest shrinks on its apical surface. Decreasing growth of LECs reduces the pressure leading to a final state with a larger nest. Increasing cell growth of LECs increases the pressure leading to a final state with a narrower nest. Stiffening the nest through the introduction of cell divisions leads to a final state with a larger nest.

epidermis. Increased mechanical compliance of histoblasts, possibly aided by their basal growth, would then lead to the narrowing of the histoblast nest. Perturbation experiments support this working hypothesis: (i) reducing the growth of LECs leads to wider nests; (ii) increasing the growth of LECs leads to narrower nests; and (iii) stiffening nests by enforcing divisions leads to wider nests.

To summarize, the driving force for junctional buckling in histoblast is the imbalance between the growth of cell–cell junctions and the limited apical area of histoblasts. A change in this balance can lead to very different cell shapes in histoblasts: Increasing the surface keeping junctions unchanged (through growth reduction of LECs) or blocking the growth of junctions reduces the extent of buckling (cartoon at the bottom of Fig 8A). An important characteristic of the mechanics of histobasts is also their propensity to store bending energy on longer times scales than embryonic cells, for example. However, the stress eventually dissipates and the deformations are stabilized.

Why do the junctions of histoblasts buckle and not those of LECs? A first important aspect is the loss of the squamous nature of histoblast through the transition. The ability of histoblasts to expand basally makes them more compliant to the growth-induced pressure than LECs, which remain squamous. This compliance should facilitate the apical shrinkage. Junctions may also be less rigid in histoblasts than in LECs, which would promote buckling in histoblasts at lower mechanical loads. In line with this, LEC–histoblast junctions (at the periphery of the nest) start to buckle at the later stages of the transtion long after histoblast–histoblast junctions buckled but before LEC–LEC even started to do so (see, for example, Fig 1D). These heterogeneous junctions are expected to have intermediate stiffness and, thus, intermediate buckling behavior. The obsevation of a partial cytoskeletal depletion from histoblast-junctions is also in line with this hypothesis, although a more systematic investigation of the link between the cytoskeleton and the stiffness of junctions must be established. Most importantly, we do observe through ablation that there is some remnant tension in LECs, which probably contributes to keeping straight junctions. We observed that buckling proceeds in a plane parallel to the surface of the epithelium, even though we have not investigated a possible deformation along the third dimension at the smallest scales. Such a mechanical asymmetry could stem

from the fact that adherens junctions are sandwiched by the stiff apical extracellular matrix above, and septate junctions below, making it harder to deform junctions along the apico basal direction. Note that connection to the stiff apical extracellular matrix may contribute the elastic nature of histoblasts. The buckled junctions of histoblasts are reminiscent of fluctuating junctions observed in other tissues. For example, during early dorsal closure, amnioserosa cells show wrinkled cell junctions due to fast contractions of the acto-myosin cytoskeleton [33,57–59]. The underlying mechanism, however, is different in histoblasts where compression is the driving force. Both the extent of the folds—deeper in histoblasts—and the time scales involved —hours in histoblasts against seconds to minutes in amnioserosa—point to different mechanisms at play. There are also similarities in the shape of histoblasts and the pavement cells of plant leave [46,60]. These puzzle-shaped pavement cells have convoluted shapes that also develop over a long interval and are stabilized by a plastic remodeling [46]. Although the final shape of plant pavement cells closely resembles that of histoblasts, their makeup seems to operate in reverse. Plants have a rigid cell wall that is under tension from turgor pressure. In the case of pavement cells, the formation of lobes is driven by growth restriction in a field under tension. Instead, the formation of lobes in histoblasts is due to lateral compression, which we believe is then compensated for by a plastic remodeling. With physical mechanism operating in reverse, compression versus tension, histoblasts and pavement cells are examples of stable phenotypes that differ from the usual polygonal shape and that are governed by other mechanisms than acto-myosin tension.

Is there an advantage for histoblasts to go through junctional buckling? The folding of junctions could allow the entire histoblast nest to remain compact while its constitutive cells are growing. Owing to the potential effects of compressive forces on cell physiology (reviewed in [61]), the compact and folded state may serve to protect histoblast by taming the compressive stress. Furthermore, as the apical domain of the nest remains as small as possible on the surface of the epidermis, the chance that it will be damaged accidentally in the larval life is reduced, while still maintaining histoblasts in close contact with LECs, ready to tense up in early pupal life, and later on expand and invade the surface of the abdomen [28].

## Materials and methods

### Fly stocks

*D. melanogaster* strains were grown at 25˚C except if else stated, in standard food (Nutri-Fly "German formulation"). A list of all the strains used for this study and generated for this study is listed in Table 1. For snapshot-imaging of larvae at the same stage, eggs were collected every 4 h from the culture tube. The age is thus expressed in h AEL, and the results averaged of a time window of 4 h. For specific expression in the histoblasts or larval epithelial cells, we used fly lines carrying the esg-Gal4 or e22c-Gal4 promoters, respectively. Their specificity was tested with genetically encoded fluorescence probes such as UAS-GFP and UAS-src:GFP, as reported in S9 Fig.

Specific genotype in individual figure is given in Table 2.

### Live imaging

Live imaging was performed with a custom built confocal spinning disc setup built of an inverted microscope (EclipseTi2-E, Nikon Instruments), and a spinning disc device (CSU-X1-M1, Yokogawa). Images were acquired with a 488- and a 561-nm lasers (Sapphire, Coherent) and an iXon Ultra888 EMCCD camera (Andor, Oxford Instruments). Z-stacks were acquired with a z-interval of 1 $\mu m$. Laser power and exposure were kept as low as possible for chronic imaging to reduce phototoxicity. The 2 color channels (GFP and mCherry, 488

**Table 1. Stocks.**

| Stock description | Short name | Origin |
|---|---|---|
| ;endo-Ecad:mKate(2x) / CyoGFP; | cad:mKate | Y. Bellaiche |
| ; endo-cad:GFP; hist:RFP | cad:GFP | C. Collinet |
| ; esg-Gal4 endo-cad:GFP; | esg-Gal4 cad:GFP | this study |
| ; esg-Gal4 endo-cad:GFP; Gal80ts /TM6b | esg-Gal4 cad:GFP Gal80ts | this study |
| ;esg-Gal4 endo-cad:mKate | esg-Gal4 cad:mKate | this study |
| ;e22c-Gal4 UAS-src:GFP; Gal80$_{ts}$ | e22-Gal4[TS] | |
| ;UAS-Src:GFP; | UAS-src:GFP | P. Kakanj |
| yw;;UAS-InR(DN) | UAS-InR-DN | Bloom. 8253 |
| UAS-stg.N/CyO | UAS-stg | Bloom. 4777 |
| y,w,MiPT-GFSTF.0dlg1[MI06353-GFSTF.0] | Dlg:GFP | Bloom. 59417 |
| yw, sqhAx3; sqh-sqh:GFP; sqh-sqh:GFP | sqh:GFP | Bloom. 66675 |
| y[1]w[*];; UASp-YFP.Rab11.S25N-06 | UASp-YFP.Rab11.DN | Bloom. 23261 |
| y[1] sc[*] v[1] sev[21]; Py[+t7.7] v[+t1.8] = TRiP.HMS00693attP2 | UAS-Cad-RNAi | Bloom.32904 |
| ;; UASp-affimer06:GFP | UAS-aff06:GFP | M. Mavrakis |
| ;UAS-Tsc1-RNAi; | UAS-Tsc1-RNAi | Ref. [62], gift from S. Hansarma |

Bloom. = Bloomington stock center. Short name = name used in the main text.

and 564 nm lasers, respectively), where acquired in sequence. All images were obtained with a 60X water-immersion objective (Plan Apo 60×, NA 1.2, Nikon).

**Sample preparation.** We refer to snapshot imaging when each larva was imaged just once, at a specific time. To characterize the buckling transition, we imaged staged cad: mCherry larvae at different ages (h AEL). To compare different mutants and obtain other measurements (i.e., myosin and actin content, cell volume, cell thickness), we chose larvae at the wandering stage, when the buckling transition has accomplished. Larvae were anesthetized with a custom-built chamber made with a glass-bottom Petri dish (MakTek), and a 3D-printed lid with 2 inlets (S1 Fig), connected to 2 syringes via rubber tubings (VWR, Tygon 3603). One tube can be closed by by a 2-way manual valve (Masterflex, 30600–00) and the other by a 3-way valve (Masterflex, 30600–01). About 200 $\mu l$ of Desflurane (Suprane, Baxter) were injected in 1 syringe, then the syringe closed to 5 $\mu$l and the liquid let expand to about 18 $\mu$l, by closing the valve. In the meanwhile, larvae were washed in PBS and placed in the Petri dish.

**Table 2. Genotypes.**

| Figure | Genotype |
|---|---|
| Fig 1 | ;endo-Ecad:mKate(2x) / + |
| Fig 2A–2F | ; esg-Gal4 endo-cad:mKate / UAS-Src:GFP |
| Fig 2K and 2L | Dlg:GFP / +; endo-cad:mKate / + |
| Fig 3A | ;endo-Ecad:mKate(2x) / + |
| Fig 3B | ;esgGal4 endo-Ecad:mKate(2x) / +; UAS-RNAi-Cad / +; |
| Fig 3C | ;esg-Gal4 endo-Ecad:GFP / UAS-YFP.Rab11.DN; |
| Fig 3D | ;esg-Gal4 endo-Ecad:GFP / UAS-Stg.N; |
| Fig 4 | ;endo-Ecad:GFP |
| Fig 6 | sqhAx3 / +; sqh-sqh:GFP / endo-Ecad:mKate; sqh-sqh:GFP / + |
| Fig 7A | ;endo-Ecad:mKate(2x) / + |
| Fig 7B | ;e22c-Gal4 UAS-src:GFP / endo:cad:GFP; UAS-InR-DN / Gal80$_{ts}$ |
| Fig 7C | ; e22c-Gal4 endo-cad:mkate UAS-GFP / UAS-Tsc1-RNAi; + / Gal80$_{ts}$ |

After a first anesthetization of about 5 min, the valves were closed, and the Petri dish open to allow alignment of the larvae to image histoblasts. The anesthetic was then reinjected and the larvae imaged immediately after.

Embryos were prepared following Sumi and colleagues' study for S3 Fig and Cavey and colleagues' study for S5 Fig.

**Chronic imaging.** We refer to chronic imaging when the same larvae were imaged several times, at different ages. The anesthetization and imaging protocol adapted from [63] is schematized in S1 Fig. Larvae were washed and anesthetized as described above, but with a lower anesthetic dose (150 $\mu$l). To avoid potential effects on growth due to starvation, we limited the anesthetization time (including imaging) to about 30 min. After each imaging session, larvae were put one by one in a humid chamber with soft food and incubated at 25°C. With this protocol, all analyzed larvae survived to the adult phase.

## Laser dissection experiments

Laser dissection was implemented on a home-built system described in Meng and colleagues' study [64]. The system couples a near-infrared 130 fs fiber laser (YLMO 930 ± 10 nm, Menlo-Systems) operating at 50 MHz for cell junction severing, with an inverted Nikon Eclipse Ti microscope (Nikon Instruments) equipped with a Yokogawa spinning disk unit (CSU-X1, Yokogawa Electric) to image the resulting mechanical relaxation. In the setup, imaging is controlled in Matlab via a $\mu$Manager [65] /Matlab interface. Synchronization with laser dissection, and other elements of the instrument (galvo-scan, z-scan, shutters, filter wheel), is performed with an input/output data acquisition card (USB-6251, National Instruments).

Severing of epithelial junctions was performed by tightly focusing the NIR laser in the focal plane using a water-immersion objective (Plan Apo 60×, NA 1.2, Nikon), which results in a local plasma ablation. With an average power of 154 mW at the back aperture of the objective, the focused laser delivers a pulse peak power density near $7.42 \times 10^{12}$W·$cm^{-2}$. With a pulse duration of 130 fs, the system has the prerequisite to perform plasma-induced ablation [66,67]. To sever the tissue along lines the laser beam was moved 2 to 3 times along the targeted region in the sample with the help of a galvo-based scanner (Cambridge Technologies) at a constant speed of about 500 ums⁻1.

Imaging was performed before and after dissection with the help of 488 nm diode laser (2 mW nominal, coherent OBIS LX) using the same objective lens. For all dissection experiments, an initial image stack was acquired prior to laser dissection for the user to interactively locate the severed domain. For LECs and histoblasts, time-lapse imaging was acquired at a frame rate of 5 fps to visualize severing and the resulting relaxation. To quantify the relaxation, a kymograph is generated along the severed junction using Fiji and used to measure its opening.

## Optical tweezers

Optical manipulation was done following [41] using a spinning disk microscope (Perkin-Elmer) coupled with a single-beam gradient trap (1,070-nm wavelength, ytterbium fiber laser, IPG). A 100× water-immersion lens (CFI Plan Apochromat VC 100× oil, NA 1.40, WD 0.13) was used for imaging as well as introducing the optical trap in the imaging plane. Galvanometric mirrors controlled laser trap position deflection to produce sinusoidal oscillations. The period used for the sinusoid is of the order of 2 s. Before every in vivo experiment, the relationship between galvanometer voltages and laser trap position was calibrated using fluorescent beads (localization precision of 25 nm). During experiments, both images and position of the galvanometers were simultaneously recorded. In S1 Movie, Because the signal is very noisy (due to bleaching and noise of the camera), the right panel provides a denoised version using the nlmeans approach [68] as implemented in the gmic library [69].

## Image analysis

Image processing and data analysis were mainly performed in Matlab using custom-written scripts. Preprocessing image segmentation of cad:mKate and cad:GFP projections were done in Ilastik [70] and TissueAnalyzer for ImageJ.

**Morphological analysis of histoblasts' junctions.** For morphological analysis, z-stacks of cadherin junctions obtained by confocal spinning disk microscopy were projected by simple maximum projections, when the tissue was well positioned and parallel to the glass slide, or by a curved projection when the histoblast plane was tilted relatively to the imaging plane. The surface detection algorithm and curved projection were performed using the procedures described in Abouakil and colleagues' study [71]. The maximum intensity projections were then segmented using Ilastik [70] and Tissue analyzer [72]. The segmentations were then used to calculate cell area, perimeter, and circularity. We represented each parameter and group of data as a box plot containing the mean value (diamond-shaped marker), median (middle solid line), lower and upper quartiles (box limits), outliers (dots), and minimum and maximum non-outliers values. The notches and shaded regions represent the 5% confidence interval, i.e., if 2 boxplots have superposed notches, the 2 data set are considered from the same Gaussian distribution. In InR-DN and TSC1-RNAi experiments, the area of LECs was measured using the same segmentation approach. The area was then measured in LECs within 3 cell diameters from the LECs/histoblast interface.

**Myosin and actin content analysis.** The surface-detection algorithm by Abouakil and colleagues [71] was used to define the plane of cadherin junctions. The identified surface was then used as a mask to analyze the fluorescence signal coming from junctional actin or myosin. The junctional enrichment was then calculated as the ratio of the normalized intensity of the junctions and of the cytosol, at the corresponding plane.

**Volume estimation.** cad:mkate esg>GFP flies were imaged to estimate the total nest volume, as a function of the average cell circularity. We analyzed Cad:mKate stacks as described in the morphometric analysis section. To obtain an estimate of the volume, we first equalized the intensity values of all stacks, then used Ilastik to obtain 3D segmentation of the GFP signal. Finally, we analyzed and plotted the data with Matlab, MathWorks.

**Thickness measurements.** Histoblast thickness was measured from cad:mkate esg>src:GFP flies. Cad:mKate stacks were analyzed as described above. Esg>src:GFP stacks were analyzed using a custom script coded in Matlab as follows. For each xy pixel, the z signal was fitted to a double Gaussian to identify the positions of the intensity peaks, i.e., of the apical and basal membrane at each position. For each nest, we thus calculated the average thickness as the average distance between the 2 membranes. Data from different stacks were then pooled to obtain average values as a function of cell circularity.

## Statistical analysis

For each experiment, we used a minimum of 10 larvae. In the case of TSC1-RNAi, InR-DN only 4 and 3, respectively, of them could be analysed because of poor image quality. The precise numbers for each experiment are listed hereafter. WT: 69 (11, 14, 11, 26, 7, respectively, at <90 h AEL, 90 to 95 h AEL, 95 to 100 h AEL, wandering stage, white pupa), sqh:GFP: 18, Af: GFP: 12, src:GFP: 10, Rab11-DN: 12, stg: 11. Statistical analysis and plotting were done using built-in functions in Matlab and R. First, all measured variables (area, perimeter, circularity) were tested for normality using Lillifors test (Table B in S1 Text). Then, normally distributed variables were compared by *t* test, while nonnormally distributed variables were evaluated by Mann–Whitney nonparametric test. The detailed outcome of the statistical tests relative to all measurements in the study can be found in the Supporting information. Table A in S1 Text

reports on $p$-values to test the difference of distributions presented in Fig 1. Table B in S1 Text reports on normality test for Fig 1. Table C in S1 Text reports on statistical test relative to Fig 3. Table D in S1 Text reports on statistical test relative to Fig 7. We chose the following convention for $p$-values in figure panels, *: $p<0.05$; **: $p<0.01$ ***: $p<0.001$; ****: $p<0.0001$ and lower.

## Model of histoblast shape formation

The model of histoblast cell shape formation was implemented in MorphoDynamX (www. MorphoDynamX.org), a framework for the modeling of growing tissues. This model was derived from the 2D model of cell shape development published in Sapala and colleagues' study [46] and is described below.

The model represented a 2D section through the apical surface of the histoblasts, where cell boundaries were represented as point masses (vertices) connected by linear springs (edges). Each vertex contained a position, whereas each edge was assigned a resting length representing the unstressed state of its corresponding spring. The curvature between adjoining edges was maintained by angular springs, with a resting angle representing the unstressed state of these springs. The positions of vertices on the template's boundary were fixed (i.e., Dirichlet boundary conditions). To prevent self-intersection of the cell boundaries, a penalty method was used, which applies a repulsive force on vertices that come into close proximity to each other (c.f. Parent [73]). Additionally, an elastic foundation impeding the rapid displacement of the boundary was modeled by adding springs that pull vertices back to their previous positions.

**Mechanical simulation.** To provide a mechanical representation for cells, linear and angular springs were employed. Sections of cell boundary in the simulation act as linear springs, with forces ($F$) proportional to their stretch:

$$F = k_b \left( \frac{l}{l_{rest}} - 1 \right),\qquad(1)$$

where $l$ is the length of the segment, $l_{rest}$ is the rest length, and $k_b$ is a spring constant controlling the stiffness of the spring. The springs representing the recoil of the elastic foundation and connections across the cell employ the same formula but use different spring constants ($k_{foundation}$ and $k_{connection}$; see Table 3). Different spring constants can be specified for compression and tension, and for the connection springs, only compression generates force. Vertices on cell boundaries also employ angular springs to create bending forces proportional to the difference between the angle between their neighbors and a rest angle, as described in Sapala and colleagues' study [46].

**Table 3. Parameters for simulations of histoblast shape formation.**

| Name | Symbol | Value | Description |
|---|---|---|---|
| Subdivide threshold | $Th_{sub}$ | 1 μm | Length threshold for subdividing cell boundary segments |
| Boundary stiffness | $k_b$ | 0.2 | Stiffness of linear boundary springs |
| Bending stiffness | $k_{bend}$ | 0.0005 | Stiffness of angular boundary springs |
| Connection stiffness | $k_{connect}$ | 0.1 | Stiffness of connections across cells |
| Template growth | $(g_X, g_Y)$ | (0, −0.6) | Growth rate of the template in (x,y) directions |
| Boundary growth | $g_b$ | 1.0 | Growth rate for cell boundaries |
| Growth time step | $dt$ | 0.005 | Time step for growth |
| Foundation stiffness | $k_{foundation}$ | 0.1 | Spring constant for elastic foundation |
| Foundation time step | $dt_{foundation}$ | 0.01 | Relaxation time step for the elastic foundation |

**Simulation initialization.** The simulation was initialized with an irregular polygon grid that approximated the size and shape of histoblast cells prior to the emergence of puzzle-like forms. To obtain visually plausible cell shapes, we used the CellMaker tool in MorphoDynamX. This tool successively divided an initial rectangular cell according to the shortest wall rule, with pinching applied to the dividing cell (shortening the new wall; see Smith and colleagues [74]). A patch of cells was then selected for the simulation. Prior to simulation, boundary segments exceeding a threshold length $Th_{sub}$ were subdivided. Finally, the rest-length for each wall-segment was initialized to its current length, whereas the rest angles at each vertex were set set to $2\pi/n$ where $n$ was the number of neighbors for the vertex.

**Simulation loop.** Once the model was initialized, simulations proceeded according to the following simulation loop. At the start of each iteration, connections are added across the cell to prevent self-intersection via a spring-based penalty method. Growth is then applied by increasing the rest length of the springs on the cell boundaries, while reducing the size of the simulation domain along the X direction to emulate compression by the surrounding larval epithelial cells. Vertices are then moved to restore mechanical equilibrium using the backward Euler implicit solver provided by MorphoDynamX. At the end of the simulation loop, once mechanical equilibrium is found, cell-wall segments exceeding the threshold $Th_{sub}$ were subdivided to maintain a smooth approximation of the cell-wall, and the positions of the elastic foundation are moved towards the current vertex positions.

**Connection placement.** To prevent the intersection of cells as the template shrinks and the cell boundaries grow, connections (springs) were added across cells connecting vertices that are positioned within a threshold length $Th_{sub}$ and are visible to each other. This creates a spring-based force that introduces an energetic penalty for self-intersection that helps to prevent and resolve boundary–boundary collisions (also called a penalty method; see Parent [73]). To have a connection, vertices must be at least 2 vertices away (i.e., they cannot be neighbors or neighbors of neighbors). The connections are adjusted in length depending on the angle with the cell boundary and are only made if this angle is over a threshold (i.e., are within some angle to perpendicular of the cell boundary). Lengths are set to lie on the tangent circle of diameter $Th_{sub}$. These connections are reset every iteration of the simulation and only resist compression. Specifically, if the rest length is shorter than the actual length they induce no force. A length factor is included to allow the creations of connections that are longer than $Th_{sub}$, although above this length, they create no force. These serve to adapt to movements in positions that may occur during solving for mechanical equilibrium, so that connections are not missed if vertices move closer together during a solving step.

**Growth and domain shrinkage.** During each time step, the cell boundary edges were grown by increasing their rest length $l_{rest}$ as follows:

$$l_{rest} = (1 + dt \cdot g_b)l_{rest}, \tag{2}$$

which approximates exponential growth of the boundary according to the time step $dt$ and growth rate $g_b$. Conversely, the template shrank exponentially over time by scaling X-positions and Y-positions vertices by $1+g_X \cdot dt$ and $1+g_Y \cdot dt$, respectively, during each step of the simulation. To achieve contraction along the X direction, we set $g_X < 0$ and $g_Y = 0$. Shrinking the domain further increased the density of cell boundaries in the template.

**Elastic foundation.** An elastic foundation was created by generating a foundation position $P_{foundation}$ for each vertex connected to the vertex's current position $P_{current}$ by a linear spring with stiffness $k_{foundation}$. During each time step, the foundation position is updated by

moving it towards the current position according to the following equation:

$$P_{foundation} = P_{foundation} + (P_{current} - P_{foundation})dt_{foundation}. \tag{3}$$

The effect of the foundation is controlled by the strength of the foundation springs $k_{foundation}$ and speed $dt_{foundation}$ at which the foundation positions are moved towards their actual positions.

**Parameters.**   Simulation parameters for the model are given in Table 3.

## Supporting information

**S1 Text. Supporting tables.** Table A. Statistical tests for pair-wise comparison of the distributions presented in Fig 1. Table B. Normality test for parameter distributions in Fig 1. Table C. Statistical tests of the results presented in Fig 3. Tables D and E. Statistical tests relative to the results shown in Fig 7. Table F. Statistical tests relative to the results shown in S7 Fig. H and *p*-values reported were calculated by Mann–Whitney U test.
(PDF)

**S1 Fig. Anesthesia chamber and imaging protocol.** (**A**) Assembled anesthesia chamber made of a glass-bottom dish (zoom in **B**), a custom-designed injection lid (zoom in **C**), tubing, and valves to which syringes are then connected. (**D)** Main steps of the imaging protocol: (1) larvae are washed in PBS and dried on a lab wipe; (2) larvae are positioned around the glass of the Petri dish and anesthetized for 5 min; (3) after closing the valves to keep the anesthetic, larvae are aligned on the glass, with a drop of halo-carbon oil; 4) larvae are images by confocal spinning disc microscopy; and (5) for chronic imaging, each larva is placed in a Petri dish with soft food and let recover for a few hours before repeating the procedure.
(TIF)

**S2 Fig. Comparison of different histoblast nests at the wandering stage. Upper panel**: anterior dorsal, posterior dorsal, and ventral histoblast nests from the sixth segment of the same larva at the wandering stage are shown. Histoblasts from all nests show the same buckling phenotype. We chose to image the anterior dorsal nests because of 2 main reasons: they are formed by a higher cell number, and they are flatter than the ventral nests, which are often located close to the limit between 2 abdominal segments. **Middle panel**: abdominal nest from the same larva, which features a deep fold due to the location close to the segment limit. This makes the image quality low, and the analysis less reliable. **Lower panel**: anterior dorsal nests from 1 to 6 of the same wandering larva. Scale bars = 20 μm.
(TIF)

**S3 Fig. Comparing movements in embryonic amnioserosa and larval histoblasts nests.** Ten trajectories of vertices in the amnioserosa at the onset of dorsal closure are show on the left. Ten trajectories of vertices of histoblast from wandering stage larva are shown on the right. Thirty points were measured with a time interval of 1 min. In vertices from the amnioserosa, a global drift due to morphogenetic movement was removed by subtracting the displacement of the center of mass of all vertices.
(TIF)

**S4 Fig. 3D characterization of histoblast growth.** (**A**) Z-projection of cad:mKate (magenta) and cytosolic GFP (cyan) taken in a larva at 90 h AEL, i.e., at the beginning of the buckling transition. (**B**) Z-projections of the same larva 20 h later, after the formation of junctional lobules. (**C, D**) Masks of the external contour of the apical and basal sides of the larva in (**A**). The white area is obtained from the apical plane obtained from cad:mkate (z1 in the schematic

representation), the gray area from the cytosolic GFP maximum projections. It corresponds to the basal plane at which the nest is the largest (z2 in the schematic representations). Before the transition (**C**) z1 and z2 are almost superposed, while after transition (**D**) z2 is much bigger than z1, meaning that HBs expand basally. (**E**) Ratio of basal/apical area for different circularity values, obtained from the masks of the z-projections as shown in C and D, i.e., z2/z1. As HBs junctions fold, the basal areal becomes larger than the apical. Apical and basal areas correspond to the the adherens region and the largest basal area, as schematized below the plot. (**F**) Total nest volume at different times. White, gray, and black dots correspond each to one histoblasts nest. As apical surface shrink, histoblasts expand below the adherens region, as schematized below the plot. The data underlying the graphs shown in the figure can be found in S1 Data.
(TIF)

**S5 Fig. Mechanical stimulation of an individual junction in the early *Drosophila* embryo.** (**A**) An optical tweezer is used to oscillate a junction (point 1 on the image) in the germ band at the onset of germ band extension. (**B**) Kymographs display an absence of oscillations at different points around the stimulation.
(TIF)

**S6 Fig. Junctional enrichment of actin.** (**A-D**) Live imaging of cadherin and actin at the level of junctions at 90 h AEL (**A**, **B**) and 110 h AEL (**C**, **D**). ZX and ZY orthogonal views in (**D**) signal show the actin enrichment at the tricellular interfaces along the apico-basal axis (arrows). Scale bar = 10 μm. (**E**) Plot of relative amount of junctional actin as a function of circularity, calculated as the ratio junctional signal $Act_{junct}$ over cytosolic signal $Act_{cyto}$ as represented in the schematic. Note that the axis of circularity has been inverted to reflect temporality. In the box plot, the horizontal bar represents the median for each bean, the shadowed areas the confidence interval of 0.05, the diamonds correspond to the mean value for each bin and the yellow circles are single data points. Pearson's correlation coefficients calculated on all the data where 0.18 with a *p*-value of 0.002. *t* Test comparisons for the junctional enrichment of the first and last point gave *p*-values of 0.002. For each bin, $N$ = 116, 88, 32, 27, 24. The data underlying the graphs shown in the figure can be found in S1 Data.
(TIF)

**S7 Fig. Characterisation of the epidermis in e22c>InR-DN pupae.** (**A**) Live imaging of the epidermis of a WT white pupa. (**B**) Live imaging of the epidermis of a white pupa in which LECs growth was reduced by impairing the insulin receptor pathway, causing reduced load on histoblast junctions, hence reduced junctional buckling as compared to the WT. Yellow arrows in both indicate the width of LECs, which increases in TSC1-RNAi and decreases in InR-DN larvae. Scale bar = 10 μm. (**C, D**) Box plots showing the quantifications of histoblasts area in WT and InRdn white pupae. *p*-Values were calculated by Mann–Whitney U test. $N$ = 74 (WT), $N$ = 38 (InRdn) cells. The data underlying the graphs shown in the figure can be found in S1 Data.
(TIF)

**S8 Fig. Characterisation of the LECs' size in e22c>InR-DN and e22c>TSC1-RNAi larvae.** (**A**) Live imaging of the LECs of a WT wandering stage larva. (**B**) Live imaging of the LECs in which growth was reduced by impairing the insulin receptor pathway. (**C**) Live imaging of the LECs in which growth was increased by expressing TSC1-RNAi. The schemes above the images represent cell growth in larval epidermis. In the WT, LECs (in grey) and histoblasts (light blue) are growing at their physiological rate, with LECs faster than hb. This is indicated by 1 plus sign + for hb, and 2 (++) for LECs. In InR-DN larvae, LECs' growth is slowed down,

represented by a single + in the scheme. On the contrary, in TSC1-RNAi, LECs' growth is increased, as represented by the 3 plus signs, +++. Scale bar = 10 μm.
(TIF)

**S9 Fig. Specificity of the esg-Gal4 and e22c-Gal4 promoters. Left:** Expression of cytosolic GFP under the Esg-Gal4 promoter. The image shows that the promoter of the escargot gene is specific to histoblasts and does not express in the surrounding LECs. The anterior and posterior dorsal nests are visible. Scale bar = 20 μm. **Right:** Expression of src:GFP under the e22c-Gal4 promoter. The image shows that the driver expresses in LECs, but not in histoblasts. The anterior nests is recognisable as a dark area. Scale bar = 20 μm.
(TIF)

**S1 Movie. Movie corresponding to Fig 4E.** The sequence is acquired while the point along a junction indicated by an arrow is horizontally stimulated with an optical tweezer. The left panel corresponds to the raw data. Because the signal is very noisy (due to bleaching and noise of the camera), the right panel provides a denoised version using the nlmeans approach [68] as implemented in the gmic library [69]. The total frame is approximately $10 \mu m$.
(MP4)

**S2 Movie. Movie corresponding to S5 Fig.** The sequence is acquired while the point along a junction indicated by an arrow is horizontally stimulated with an optical tweezer. The total frame is approximately $15 \mu m$.
(MP4)

**S1 Data. The data underlying the graphs shown in Figs 1–4, 6, 7, and S4–S7.**
(ODS)

# Acknowledgments

We thank Manos Mavrakis, Sophie Brasselet, Raphael Clément, and Martine Ben Amar for fruitful discussions on this project. We thank Frédéric Galland, Philippe Roudot, and Nicolas Desprat for advices on image analysis. We thank Yohanns Bellaiche, Jérome Solon, Manos Mavrakis, Claudio Collinet, and Parisa Kakanj for sharing stocks.

# Author Contributions

**Conceptualization:** Annafrancesca Rigato, Loïc LeGoff.

**Data curation:** Annafrancesca Rigato, Loïc LeGoff.

**Formal analysis:** Richard S. Smith.

**Funding acquisition:** Loïc LeGoff.

**Investigation:** Annafrancesca Rigato, Loïc LeGoff.

**Methodology:** Huicheng Meng, Claire Chardes.

**Project administration:** Loïc LeGoff.

**Resources:** Huicheng Meng, Claire Chardes, Adam Runions, Faris Abouakil, Richard S. Smith.

**Software:** Adam Runions, Faris Abouakil, Richard S. Smith.

**Supervision:** Loïc LeGoff.

**Visualization:** Annafrancesca Rigato, Richard S. Smith.

**Writing – original draft:** Annafrancesca Rigato, Loïc LeGoff.

**Writing – review & editing:** Richard S. Smith, Loïc LeGoff.

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
