## [Editor Report · Decision Letter 0]

4 Apr 2023

Dear Dr Le Goff, 

Thank you for submitting via Review Commons the revision of your manuscript entitled "Morphogenesis of Drosophila Larval Histoblasts Proceeds through Buckling of their Apical Junctions" for consideration as a Research Article by PLOS Biology.

Your manuscript has now been evaluated by the PLOS Biology editorial staff as well as by an academic editor with relevant expertise and I am writing to let you know that we would like to send your submission back to the original reviewers.

Once your full submission is complete, your paper will undergo a series of checks in preparation for peer review. After your manuscript has passed the checks it will be sent back to the reviewers. To provide the metadata for your submission, please Login to Editorial Manager (https://www.editorialmanager.com/pbiology) within two working days, i.e. by Apr 06 2023 11:59PM.

Kind regards,

Ines

--

Ines Alvarez-Garcia, PhD

Senior Editor

PLOS Biology

---

## [Decision Letter · Decision Letter 1]

13 Jun 2023

Dear Dr Le Goff,

Thank you for your patience while we considered your revised manuscript from Review Commons entitled "Morphogenesis of Drosophila larval Histoblasts Proceeds through Buckling of their Apical Junctions" for publication as a Research Article at PLOS Biology. Please also accept my apologies for the delay in providing you with our decision. Your revised study has been evaluated by the PLOS Biology editors, the Academic Editor and by the two original reviewers.

The reviews are attached below. You will see that both reviewers agree that the manuscript has been improved in the revision, but they both raise several issues that remain to be addressed. Reviewer 1 thinks you should test whether increased LEC growth results in reduced histoblasts area and to provide some missing controls. Reviewer 2 thinks that the mechanisms undertying the behaviour of the bucking junctions is not sufficiently explored and suggests performing live imaging to explore the properties in more detail.

After discussing the reviews with the Academic Editor, we would like to invite you to address the remaining points raised by the reviewers. However, regarding some of Reviewer 2’s comments, the academic editor is not sure how live imaging will address flexibility, thus we won’t make this a requirement for publication. In addition, while some of the experiments suggested on LEC growth should be considered - LEC ablation, analysis of cell and nest area – the suggested detailed study of the morphology of LECs and histoblasts by live imaging at the start of buckling would not be necessary to be included in the revision.

Given the extent of revision needed, we cannot make a decision about publication until we have seen the revised manuscript and your response to the reviewers' comments. Your revised manuscript is likely to be sent for further evaluation by all or a subset of the reviewers.

**IMPORTANT - SUBMITTING YOUR REVISION**

3. Resubmission Checklist

a) *PLOS Data Policy*

b) *Published Peer Review*

Sincerely,

Ines

--

Ines Alvarez-Garcia, PhD

Senior Editor

PLOS Biology

Reviewers' comments

Rev. 1:

The authors have satisfactorily addressed most of my comments and the manuscript is much improved. I have some remaining comments:

Fig. 3B. The authors claim that cell junctions are straighter when E-cadherin is depleted. However, the cell junctions (labelled by E-cad::mKate) are difficult to see. Thus, it is difficult to assess whether Cad-RNAi results in straighter junctions (the authors are also unable to segment the images for shape quantification (line 136)). The authors should use a different marker for cell junctions (or plasma membrane) which is not affected by E-cadherin depletion.

Fig. 3E. The authors should state in the legend what the color code refers to.

Lines 195-208. The authors refer to Fig. 8E-H; Fig. 5 would probably be correct.

Fig. 7C. The authors should state which marker (cad…) they have used.

Fig. 7C. The authors show that increased LEC growth (TSC-RNAi) results in reduced histoblast area. To further test their hypothesis, the authors should determine whether in this situation also the buckling of cell junctions (circularity) is affected (like the authors do for the reduced LEC growth (Fig. S5)).

Rev. 2:

As I stated in my first review, the observation of histoblast junctional buckling is very interesting. Also, the authors' hypothesis, presented at the end of the discussion, that histoblasts buckle to ensure that they are not accidentally pushed out of the tissue/die during larval growth is intriguing.

Overall, the authors have sought to address all my comments and have improved the manuscript. As part of the revision, the authors have adapted their line of argument. Based on their data, the authors formulate the hypothesis that junctional lengthening combined with a reduction in histoblast area leads to junctional buckling. They then hypothesise that pressure by the growing LECs ultimately could cause histoblast junctional buckling.

Although the revision has improved the manuscript, two main points remain, which in my opinion need to be addressed further before the manuscript can be published in PLoS Biology.

1) The manuscript does not sufficiently explore the mechanism behind the behaviour of the buckling junctions. How are the forces created that lead to junctional lengthening, and what stabilises the buckled junctions? Laser ablation experiments show that there is hardly any relaxation when junctions of buckled histoblasts are cut, which the authors interpret as a 'plastic process [that] dissipates the compressive stress'. What is this process? In addition, the model includes 'an elastic fabric impeding the rapid displacement of the boundary'. What is this 'elastic fabric'?

Do these observations suggest that the junctions, although they are strongly bent, are very stiff/rigid? Would this depend on a rigid cortical actomyosin network? Here, the actin and myosin data seem counterintuitive. This highlights that the role of the described changes in actomyosin is not sufficiently clear.

To explore the properties of the buckling junctions in more detail, live imaging would be useful. From the authors' data, one would expect that the buckled junctions are very stable and do not change much over time (in contrast to dynamic changes in the buckling morphology, which one would expect if one would deform a flexible, already buckled rod).

Such live imaging would also help to illustrate the statement made in line 74: 'No fluctuations of the cell junctions are observed in a period of minutes.'.

2) The manuscript still does not provide sufficient support for the hypothesis that pressure from the LECs is causing histoblast buckling.

One experiment that could expand on the InR-DN and TSC1-RNAi experiments would be to ablate one LEC neighbouring the histoblast nest to release tension and ask if and how junctional morphology of the histoblasts changes.

For the InR-DN and TSC1-RNAi experiments, it would be helpful, if the authors explained in more detail which LECs were assessed for their cell area (all of them, those neighbouring the histoblasts?). If all LECs of the abdomen got bigger or smaller, would this not have strong effects on epithelial morphology, as a large area is either created or lost (or are the larvae smaller/larger)? Are LEC numbers in one segment similar in InR-DN, TSC1-RNAi and control experiments?

Also, it would be useful to know not just the cell area of individual histoblasts, but also the overall area of the nests in these experiments. This would indicate the area available for the histoblasts and further highlight the change in available space for the histoblasts. (This would also be interesting for the wild-type analysis in Fig. 1).

Also, does junctional tension change in inR-DN and TSC1-RNAi LECs?

Could a detailed study of the morphology of LECs and histoblasts by live imaging at the start of buckling help to explore whether LEC pushing is involved? For instance, does buckling begin in certain areas of the nest, which maybe correlate with specific LECs getting larger? Also, the authors discuss that buckling begins later at the LEC/histoblast interface - is this because of more rigid junctional tension or because LECs actively push?

It would also be interesting to explore the changes in LEC area in the whole hemisegment. Are all LECs growing? Is the change in LEC shapes consistent with a potential increase in pressure on histoblast nests?

Further points:

I wonder whether Fig. 6 should come before Fig. 5, because the authors talk about actomyosin at the junctions when they discuss Fig. 4 (line 177 'This could proceed through a change in the structure or composition of actomyosin in the junctions.').

The authors could show a quantification of cell shapes for the experiments in larvae presented in Fig. 7. They only show the quantification for pupae (in Fig. S5).

Also, a reference to Fig. S5 is missing in the main text.

The authors should include the number of histoblasts for each experiment to make clear that the phenotypes are not due to a reduced cell number. For some experiments, the authors state these data in the response to the reviewers, but the data are missing in the manuscript.

It would also be useful to know how many larvae/pupae were analysed for each experiment. Currently, only the n-numbers for the number of analysed cells are given in the manuscript.

In Fig. S4, the n-numbers are highly variable. In particular, there is only a single data point for a circularity of 0.7 - this n-number needs to be increased.

Paragraph starting at line 196: figure numbering is incorrect here.

Line 349: a figure reference is missing.

Fig. S3 legend: There is an incomplete sentence - 'E Ratio of basal/apical area for different circularity values, obtained from .'.

Fig. S5: The yellow arrows could be explained.

Fig. S6: The schemes above the images could be explained in more detail - they are not fully clear without consulting the other figures that have these schemes.

Fig. S7, right image: This is interesting. It would be useful to see a lower-magnification image of the whole segment, because, in pupae, en.Gal4 only drives expression in the posterior compartment (in both LECs and histoblasts). Also, which nests are shown? In pupae, I would expect en.Gal4 expression in the dorsal posterior histoblast nest.

---

## [Decision Letter · Decision Letter 2]

22 Mar 2024

Dear Dr Le Goff,

Thank you for your patience while we considered your revised manuscript entitled "Morphogenesis of Drosophila larval Histoblasts Proceeds through Buckling of their Apical Junctions" for publication as a Research Article at PLOS Biology. This revised version of your manuscript has been evaluated by the PLOS Biology editors, the Academic Editor and the two original reviewers.

The reviews are attached below. Based on the reviews, we are likely to accept this manuscript for publication, provided you satisfactorily address the remaining points raised by Reviewer 2. Please also make sure to address the data and other policy-related requests stated below.

In addition, we would like you to consider a suggestion to improve the title:

"The folded morphology of larval Drosophila epidermis is achieved by buckling of histoblast apical junctions"

We expect to receive your revised manuscript within two weeks. 

*Published Peer Review History*

*Press*

Sincerely,

Ines

--

Ines Alvarez-Garcia, PhD

Senior Editor

PLOS Biology

Fig. 1E-I; Fig. 2I; Fig. 3F-I; Fig. 4D; Fig. 6E; Fig. 7D, E; Fig. S4E, F; Fig. S6E and Fig. S7C, D

CODE POLICY

Per journal policy, as the code that you have generated is important to support the conclusions of your manuscript, we require that you make it publicly available without restrictions upon publication. Please ensure that the code is sufficiently well documented and reusable, and that your Data Statement in the Editorial Manager submission system accurately describes where your code can be found. For example, you can deposit it in Zenodo and obtain a DOI.

Reviewers' comments

Rev. 1:

The authors have satisfactorily addressed my comments.

Rev. 2:

The authors have addressed my comments and I recommend publication of the manuscript.

I very much appreciate the amount of detail given in the response to the reviewer. Also, the changes in the text helped to clarify the cell mechanics argument presented in the manuscript.

A few minor points:

I did not explain one of my 'further points' very well. In the fourth point, I meant to say that I wonder whether it would be a good idea to state the number of histoblasts per nest for the experiments shown in Figs. 3A,B,D,E and 7A,B,C. This would allow the reader to assess whether the number of histoblasts is affected by the manipulation. A change in histoblast number might change the mechanics of the system.

The authors explain in the response to the reviewer how they measured LEC area in the InR-DN and TSC1-RNAi experiments ('We measured the area of LECs within 3 cell diameters of histoblast nests…'). It would be helpful to have this information in the Methods section.

The movies do not play in Quicktime. Maybe the authors could try a different codec?

Would it be helpful to show the embryo data (Fig. S5b) next to the histoblast data in Fig. 4 to allow an easier comparison?

Drosophila should be in italics.

Line 37 - 'Histoblasts do not exchange neighbors either during this period' has no reference.

Line 50 - typo, 'drosophila' should be with a capital letter

Line 160 - the abbreviation for string could be introduced here (stg).

Line 216 - typo 'though'

---

## [Editor Report · Decision Letter 3]

3 May 2024

Dear Dr Le Goff,

Thank you for the submission of your revised Research Article entitled "A mechanical transition from tension to buckling underlies the jigsaw puzzle shape morphogenesis of histoblasts in the Drosophila epidermis" for publication in PLOS Biology. On behalf of my colleagues and the Academic Editor, François Schweisguth, I am delighted to let you know that we can in principle accept your manuscript for publication, provided you address any remaining formatting and reporting issues. These will be detailed in an email you should receive within 2-3 business days from our colleagues in the journal operations team; no action is required from you until then. Please note that we will not be able to formally accept your manuscript and schedule it for publication until you have completed any requested changes.

PRESS

Sincerely, 

Ines

--

Ines Alvarez-Garcia, PhD

Senior Editor

PLOS Biology
